# eOptShrinkQ: Near-Lossless KV Cache Compression Through Optimal Spectral Denoising and Quantization

## Abstract

We show that the key-value (KV) cache in transformer attention heads admits a natural decomposition into a low-rank *shared context* component and a full-rank *per-token* residual, well described by the spiked random matrix model. This observation leads to eOptShrinkQ, a two-stage compression pipeline: optimal singular value shrinkage (eOptShrink) automatically extracts the shared structure, and the residual—which satisfies the *thin shell property* with delocalized coordinates—is quantized by TurboQuant (40), a recently proposed per-vector scalar quantizer with near-optimal distortion guarantees. By restoring the isotropy that scalar quantization assumes, spectral denoising eliminates the need for both outlier handling and dedicated inner product bias correction, freeing those bits for improved reconstruction.

The theoretical grounding in random matrix theory provides three guarantees: automatic rank selection via the BBP phase transition, provably near-zero inner product bias on the residual, and coordinate delocalization ensuring near-optimal quantization distortion. Experimentally, we validate eOptShrinkQ on Llama-3.1-8B and Ministral-8B across three levels: per-head MSE and inner product fidelity, where eOptShrinkQ saves nearly one bit per entry over TurboQuant at equivalent quality; end-to-end on LongBench (16 tasks), where eOptShrinkQ at ~2.2 bits per entry outperforms TurboQuant at 3.0 bits; and multi-needle retrieval, where eOptShrinkQ at 2.2 bits closely matches or exceeds uncompressed FP16, suggesting that spectral denoising can act as a beneficial regularizer for retrieval-intensive tasks.

## 1 Introduction

**The KV cache bottleneck.** The key-value (KV) cache is a critical memory bottleneck in autoregressive inference with large language models (LLMs). For a model with $L$ layers, $h$ attention heads, model dimension $d_{\text{model}}$, head dimension $d = d_{\text{model}}/h$, and context length $T$, the KV cache stores $2LhTd$ entries in half-precision, consuming gigabytes of GPU memory at long context lengths. Reducing this footprint through quantization has emerged as a practical necessity for long-context deployment. Since attention scores are computed as inner products $\langle q, k \rangle / \sqrt{d}$, the fidelity of these inner products after compression directly determines output quality—making inner product (IP) bias and variance the key metrics for KV cache quantization.

**Per-vector scalar quantization.** Recent work has made remarkable progress on per-vector quantization of high-dimensional vectors. TurboQuant (40) achieves near-optimal distortion rates by compressing each key or value vector independently: a random orthogonal rotation isotropizes the vector, after which per-coordinate Lloyd-Max scalar quantization (25; 27) achieves near-optimal MSE for the resulting approximately Gaussian coordinates. At 3.5 bits per entry, TurboQuant matches full-precision performance on LongBench (3) with Llama-3.1-8B-Instruct—a strong baseline that we adopt as our downstream quantizer.

However, per-vector methods treat each vector independently, ignoring the *structured* nature of the KV cache. Within an attention head, a block of $n$ consecutive key or value vectors $\widetilde{S} \in \mathbb{R}^{n \times d}$ is not a collection

of independent random vectors—it contains a low-rank component reflecting shared structure across tokens. This shared structure means the quantizer's theoretical assumptions (isotropy on the unit sphere) are not fully satisfied, leading to inner product bias. TurboQuant's QJL correction addresses this at the cost of one bit per coordinate; our approach instead removes the structure before quantization, restoring the isotropy that makes per-vector quantization optimal.

**KV cache blocks as spiked random matrices.** We argue that the spectral structure of KV cache blocks is naturally described by the *spiked matrix model* from random matrix theory (4; 5). Consider how keys are generated in a single attention head: for token $t$ with hidden state $h_t \in \mathbb{R}^{d_{\mathrm{model}}}$, the key vector is $\widetilde{s}_t = \mathrm{RoPE}(h_t W_K, t) \in \mathbb{R}^d$, where $W_K \in \mathbb{R}^{d_{\mathrm{model}} \times d}$ is the learned key projection and RoPE applies a position-dependent rotation. A block of $n$ consecutive key vectors (the block size) forms the rows of $\widetilde{S} \in \mathbb{R}^{n \times d}$.

Each hidden state $h_t$ carries two types of information: (1) *local context* $\bar{h}_t$—the component that lies in a low-dimensional subspace shared by neighboring tokens within the block, reflecting common topic, discourse structure, and semantic context, and (2) *token-specific content* $\epsilon_t$—the unique semantic contribution of that particular token. Writing $h_t = \bar{h}_t + \epsilon_t$, each row of the key block decomposes as:

$$\widetilde{S}[t,:] = \underbrace{R(t) \cdot \bar{h}_t W_K}_{S[t,:]} + \underbrace{R(t) \cdot \epsilon_t W_K}_{Z[t,:]}, \tag{1}$$

where $R(t)$ is the orthogonal RoPE rotation at position $t$. This gives the spiked model $\widetilde{S} = S + Z$, where the $t$-th row of the signal matrix is $S[t,:] = R(t) \cdot \bar{h}_t W_K$ and the $t$-th row of the residual is $Z[t,:] = R(t) \cdot \epsilon_t W_K$. The signal $S$ is low-rank because the local context vectors $\{\bar{h}_t\}_{t=1}^n$ span a low-dimensional subspace of $\mathbb{R}^{d_{\mathrm{model}}}$—the rank $r$ reflects how many independent contextual directions are present within the block. Both $W_K$ (a deterministic learned projection) and $R(t)$ (an orthogonal rotation) are applied identically to $\bar{h}_t$ and $\epsilon_t$. Since deterministic linear transformations preserve the independence of their inputs, $S$ and $Z$ are independent whenever $\bar{h}_t \perp \epsilon_t$—regardless of $W_K$ or $R(t)$.

- The **signal** $S = \sum_{i=1}^r d_i \mathbf{u}_i \mathbf{v}_i^\top$ is low-rank because the local context vectors $\{\bar{h}_t\}$ span a low-dimensional subspace when projected into head space by $W_K$. The right singular vectors $\mathbf{v}_i \in \mathbb{R}^d$ encode the principal directions of this local subspace—directions that the model has learned to use for attention matching. The left singular vectors $\mathbf{u}_i \in \mathbb{R}^n$ describe how each token's participation in these shared directions varies across the block; for keys, RoPE imposes smooth positional modulation on $\mathbf{u}_i$, which predicts that consecutive tokens should share stronger low-rank structure than randomly ordered tokens. Because $S$ is low-rank, it can be stored compactly via its SVD factors rather than redundantly encoded into every row by a per-vector quantizer.

- The **residual** $Z \in \mathbb{R}^{n \times d}$ arises from the per-token variations $\epsilon_t$ projected into head space by $W_K$. We emphasize that $Z$ is *not* noise in the information-theoretic sense: it carries the token-specific information essential for distinguishing individual vectors during attention. However, from the perspective of the spiked model, $Z$ plays the role of the "noise" matrix because it lacks the low-rank structure that can be efficiently compressed via SVD. The residual exhibits a *separable covariance structure* $Z = A^{1/2} X B^{1/2}$ (38; 9), where $X \in \mathbb{R}^{n \times d}$ has independent entries representing the truly token-specific and dimension-specific variation, $A \in \mathbb{R}^{n \times n}$ captures temporal dependence among the per-token variations (nearby tokens may carry related but distinct unique information due to local context), and $B \in \mathbb{R}^{d \times d}$ captures coordinate-wise covariance of the per-token variations in head space.

The independence of $S$ and $Z$ follows from the independence of their sources: $S$ is driven by the local context $\bar{h}_t$ and $Z$ by the per-token variations $\epsilon_t$, with $\bar{h}_t \perp \epsilon_t$. As established above, $W_K$ and $R(t)$ are deterministic transformations that preserve this independence. We validate the spiked model empirically in Section 4.1, where the predicted spectral behavior (outlier eigenvalues, bulk distribution, and phase transition) closely matches observed KV cache spectra across layers and heads.

For **value** vectors, the same decomposition applies with $W_V$ replacing $W_K$ and no RoPE rotation: $\widetilde{S}[t,:] = \bar{h}_t W_V + \epsilon_t W_V$. The shared structure $S$ is still present (tokens in a block share semantic context), and the per-token component $Z$ captures what each token contributes to the attention output.

In the spiked model, a phase transition occurs at a critical strength $\alpha$ (4): shared components with $d_i > \alpha$ produce outlier singular values that separate from the bulk spectrum and are recoverable, while weaker components with $d_i \leq \alpha$ are entangled with the token-specific part and cannot be separated by any method. The threshold $\alpha$ depends on the full covariance structure through $A$ and $B$, not just the variance of $Z$—this is why the colored noise extension of optimal shrinkage is essential.

**Optimal shrinkage for structured denoising.** The classical approach to recovering $S$ from $\widetilde{S}$ via truncated SVD is suboptimal: hard thresholding discards weak-but-detectable signals and retains noise-inflated singular values. *Optimal singular value shrinkage* (10; 11; 28) addresses both issues by computing the Frobenius-norm-optimal correction to each singular value, accounting for the aspect ratio and noise distribution. Prior work (33) developed eOptShrink, which extends the OptShrink framework (28) to handle non-square matrices and colored noise with separable covariance—both essential for KV cache blocks where $n \neq d$ and the noise statistics are non-white. eOptShrink additionally provides fully data-driven rank estimation via the BBP phase transition threshold, without requiring knowledge of the noise covariance structure.

The optimal shrinkage framework under white noise (TRAD) has found wide applications, including diffusion MRI denoising (35), fetal electrocardiogram (fECG) extraction from trans-abdominal maternal ECG (32), seismic data enhancement (2), and stimulation artifact removal from intracranial EEG (1). Prior work (33) developed eOptShrink to handle the more realistic setting of colored noise with separable covariance. There, the motivating application was fECG extraction, where the decomposition $\widetilde{S} = S + Z$ separates the dominant maternal ECG ($S$) from the fetal ECG ($Z$)—two physiologically independent signals measured through the same electrode array. Crucially, the "noise" $Z$ in that setting is itself a clinically important signal with non-white statistics, and eOptShrink successfully recovers the fetal ECG morphology even under arrhythmia, outperforming white-noise methods. The KV cache compression problem shares this structure: the residual $Z$ carries meaningful per-token information with non-trivial covariance, making eOptShrink a natural fit. The key difference is that in fECG extraction, the goal is to recover $Z$ (the fetal signal); here, we use the decomposition to improve quantization of $Z$ by first removing the low-rank component $S$.

**From denoising to compression.** The connection to quantization becomes transparent through the lens of high-dimensional probability. When the entries of $X$ are i.i.d. $\mathcal{N}(0, 1/d)$, each row $x_t \in \mathbb{R}^d$ satisfies the *thin shell property* (36): $\|x_t\|_2 \approx 1$ with high probability, and the normalized vector $x_t/\|x_t\|$ is approximately uniformly distributed on the unit sphere $\mathcal{S}^{d-1}$. Moreover, the coordinates are *delocalized*: the largest coordinate satisfies $\|x_t\|_\infty \leq C\sqrt{\log d/d}$ with high probability for an absolute constant $C > 0$, meaning no single coordinate carries disproportionate energy (36, Problem 3.26). Multiplying by the deterministic matrices $A^{1/2}$ and $B^{1/2}$ rotates and stretches this sphere into an ellipsoid but preserves both the norm concentration and the delocalization—the rows of $Z = A^{1/2} X B^{1/2}$ remain well-spread across coordinates with no outlier structure. These are precisely the conditions under which $\mathrm{TQ_{MSE}}$'s random rotation plus Lloyd-Max quantization achieves near-optimal distortion: after normalizing each row and applying a Haar rotation, the coordinates become approximately i.i.d. $\mathcal{N}(0, 1/d)$, matching the Lloyd-Max codebook.

The problem is the signal $S$: its low-rank structure causes the rows of $\widetilde{S} = S + Z$ to cluster near a low-dimensional subspace, violating the isotropy that $\mathrm{TQ_{MSE}}$ requires. Our approach removes $S$ via optimal shrinkage, restoring the isotropic structure of $Z$. Let $\hat{S}$ denote the eOptShrink estimate of $S$ (Algorithm 3). The residual $R = \widetilde{S} - \hat{S}$ approximates $Z$: its spectral distribution and Frobenius norm converge to those of $Z$ (Proposition 6), and its rows inherit the thin shell and delocalization properties of $Z$ (Corollary 6). Since $\|R\|_F \approx \|Z\|_F < \|\widetilde{S}\|_F$ whenever $S$ has non-negligible energy, the inner product bias of $\mathrm{TQ_{MSE}}$—which scales with the squared norm—is reduced by a factor of $1/(1 + \mathrm{SNR}_t)$ compared to direct quantization (Proposition 6, Part 3), where $\mathrm{SNR}_t := \|S[t,:]\|^2/\|Z[t,:]\|^2$ is the per-token signal-to-noise ratio.

**Contributions.** We make the following contributions:

On the theoretical side, we provide the first formalization of KV cache blocks as spiked random matrices: local context $(\bar{h}_t)$ spans a low-dimensional subspace producing the low-rank signal $S$, per-token variation $(\epsilon_t)$ produces the residual $Z$ with separable covariance, and both are projected through the deterministic $W_K$, preserving their independence. We show that the residual $Z$, whose rows arise from i.i.d. sub-Gaussian entries transformed by deterministic covariance matrices, satisfies the *thin shell property*: its rows have concentrated norms and delocalized coordinates, $\|r_t\|_\infty \leq C\sqrt{\log d/d} \cdot \|r_t\|_2$ (36). The signal $S$ is the sole source of anisotropy that violates this property; removing it via optimal shrinkage restores the regime where per-vector scalar quantization is near-optimal, without requiring outlier handling or coordinate sub-grouping. We prove that this reduces inner product bias by a factor of $1/(1+\mathrm{SNR}_t)$ (Proposition 6), achieving near-zero bias so that all $b$ bits can be devoted to MSE reconstruction; QJL correction can optionally be added for further improvement. The rank and shrunken singular values are determined automatically by eOptShrink (33) via the BBP phase transition, with no manual tuning and no knowledge of the noise covariance structure.

On the experimental side, we validate the spiked model predictions—outlier eigenvalues, bulk distribution, and phase transition—on Llama-3.1-8B-Instruct and Ministral-8B-Instruct KV caches across all layers and heads. End-to-end on LongBench (16 tasks), eOptShrinkQ$_{\mathrm{MSE}}$ at ~2.2 bits achieves 47.4 (Llama) and 48.3 (Ministral), within 1.6 and 2.2 points of FP16 respectively—a 7× memory reduction with minimal quality loss. Even a simple rank-1 SVD at 2.04 bits matches standalone TQ$_{\mathrm{prod}}$ at 3.00 bits, demonstrating that spectral denoising is more effective than dedicating an extra bit to QJL correction.

## 2 Background

### 2.1 KV Cache in Transformer Attention

In multi-head attention, each layer computes queries $Q$, keys $K$, and values $V$ from the input hidden states. For an input sequence of length $T$ with head dimension $d$:

$$\mathrm{Attention}(Q, K, V) = \mathrm{softmax}\left(\frac{QK^\top}{\sqrt{d}}\right) V. \tag{2}$$

During autoregressive generation, the KV cache stores all previously computed key and value vectors so they need not be recomputed. The cache grows linearly with sequence length, making compression essential for long-context inference.

### 2.2 TurboQuant

TurboQuant (40) compresses each $d$-dimensional vector $x$ independently:

1. **Norm separation:** Store $r = \|x\|$ and work with $u = x/r$ on the unit sphere $\mathcal{S}^{d-1}$.

2. **Random rotation:** Apply a Haar-distributed random orthogonal matrix $\Pi$: $z = \Pi u$.

3. **Scalar quantization:** Quantize each coordinate of $z$ independently using a Lloyd-Max codebook (25; 27) optimized for the marginal distribution $\mathcal{N}(0, 1/d)$ of rotated unit sphere coordinates.

4. **Reconstruction:** $\hat{u} = \Pi^\top \hat{z}$, $\hat{x} = r\hat{u}$.

The key insight is that after norm separation, the unit vector $u = x/\|x\|$ lies on $\mathcal{S}^{d-1}$. For $u$ uniformly distributed on $\mathcal{S}^{d-1}$, each coordinate of $z = \Pi u$ follows a marginal distribution that converges to $\mathcal{N}(0, 1/d)$ as $d \to \infty$ (40, Lemma 1). The Lloyd-Max codebook, pre-computed for this Gaussian marginal, therefore achieves near-optimal scalar quantization distortion for each coordinate independently. The uniformity assumption on $u$ is critical: if the input vectors concentrate near a low-dimensional subspace rather than being uniformly spread on $\mathcal{S}^{d-1}$, the Gaussian approximation degrades and the codebook becomes suboptimal—a situation we address via spectral denoising.

We refer to this pipeline as **TurboQuant$_{\text{MSE}}$** (denoted TQ$_{\text{MSE}}$, Algorithm 1): at $b$ bits per coordinate, it minimizes the mean squared reconstruction error $\mathbb{E}[\|x - \hat{x}_{\text{MSE}}\|^2]$.

---

**Algorithm 1** TurboQuant (40): TQ$_{\text{MSE}}$ and TQ$_{\text{prod}}$

---

**Require:** Vector $x \in \mathbb{R}^d$, bit-width $b$, shared Haar rotation $\Pi \in \mathbb{R}^{d \times d}$, Lloyd-Max codebook $\mathcal{C}_b$ for $\mathcal{N}(0, 1/d)$, random projection $\Phi \in \mathbb{R}^{d \times d}$
1: **Output:** $\hat{x}_{\text{MSE}} \leftarrow \text{TQ}_{\text{MSE}}(x, b)$; optionally $\hat{x} \leftarrow \text{TQ}_{\text{prod}}(x, b)$
2: **TQ$_{\text{MSE}}$:**
3: $r \leftarrow \|x\|, \quad u \leftarrow x/r$ {norm separation}
4: $z \leftarrow \Pi u$ {random rotation}
5: $\hat{z} \leftarrow \mathcal{C}_b(z)$ {per-coordinate Lloyd-Max quantization}
6: $\hat{x}_{\text{MSE}} \leftarrow r \cdot \Pi^\top \hat{z}$ {reconstruction, $b$ bits per entry}
7: **TQ$_{\text{prod}}$ (optional, for unbiased inner products):**
8: $r_x \leftarrow x - \hat{x}_{\text{MSE}}$ {quantization residual}
9: Store: $\|r_x\|$ and $\text{sign}(\Phi r_x) \in \{-1, +1\}^d$ {+1 bit per entry}
10: For query $y$: $\langle y, \hat{x} \rangle = \langle y, \hat{x}_{\text{MSE}} \rangle + \|r_x\| \cdot \frac{\sqrt{\pi/2}}{d} \langle \Phi y, \text{sign}(\Phi r_x) \rangle$
11: **Bits per entry:** TQ$_{\text{MSE}}$: $b + 16/d$; TQ$_{\text{prod}}$: $b + 1 + 16/d$

---

TQ$_{\text{MSE}}$ minimizes mean squared reconstruction error $\mathbb{E}[\|x - \hat{x}_{\text{MSE}}\|^2]$. TQ$_{\text{prod}}$ augments it with a 1-bit QJL correction for unbiased inner product estimation:

$$\langle y, \hat{x} \rangle = \langle y, \hat{x}_{\text{MSE}} \rangle + \|r_x\| \cdot \frac{\sqrt{\pi/2}}{d} \langle \Phi y, \text{sign}(\Phi r_x) \rangle, \tag{3}$$

where $r_x = x - \hat{x}_{\text{MSE}}$ is the quantization residual and $\Phi \in \mathbb{R}^{d \times d}$ is a random projection matrix with i.i.d. $\mathcal{N}(0, 1)$ entries. The first term $\langle y, \hat{x}_{\text{MSE}} \rangle$ uses the MSE reconstruction, which is biased: the Lloyd-Max quantizer systematically shrinks each coordinate toward zero, so $\mathbb{E}[\langle y, \hat{x}_{\text{MSE}} \rangle] < \langle y, x \rangle$. The second term corrects this bias by estimating the missing inner product $\langle y, r_x \rangle$ between the query and the quantization residual. The key insight is the Johnson–Lindenstrauss property: for a random projection $\Phi$, the quantity $\frac{1}{d} \langle \Phi y, \text{sign}(\Phi r_x) \rangle$ is an unbiased estimator of $\frac{2}{\pi} \langle y, r_x/\|r_x\| \rangle$ (40), requiring only 1 sign bit per coordinate to store $\text{sign}(\Phi r_x) \in \{-1, +1\}^d$. The scaling factor $\|r_x\| \cdot \sqrt{\pi/2}/d$ converts this to an unbiased estimate of $\langle y, r_x \rangle$, yielding $\mathbb{E}[\langle y, \hat{x} \rangle] = \langle y, x \rangle$. The cost is one extra bit per coordinate and a 4× increase in MSE distortion, since only $b$ of the $b + 1$ bits are used for reconstruction.

## 2.3 Optimal Singular Value Shrinkage

Given a data matrix $\widetilde{S} = S + Z \in \mathbb{R}^{n \times d}$ where $S = \sum_{i=1}^r d_i \mathbf{u}_i \mathbf{v}_i^\top$ is a rank-$r$ signal with signal strengths $d_1 \geq \cdots \geq d_r > 0$, left and right singular vectors $\mathbf{u}_i \in \mathbb{R}^n$, $\mathbf{v}_i \in \mathbb{R}^d$, and $Z$ is a noise matrix, the goal is to recover $S$ from $\widetilde{S}$. Let $\widetilde{S} = \sum_{i=1}^{n \wedge d} \widetilde{\sigma}_i \widetilde{\boldsymbol{\xi}}_i \widetilde{\boldsymbol{\zeta}}_i^\top$ be the SVD of the observed matrix, where $\tilde{\lambda}_i := \widetilde{\sigma}_i^2$ are the eigenvalues of $\widetilde{S}\widetilde{S}^\top$. *Singular value shrinkage* (10; 11; 28) constructs an estimate of $S$ by applying a nonlinear *shrinker* $\varphi : [0, \infty) \to [0, \infty)$ to the observed singular values:

$$\hat{S}_\varphi = \sum_{i=1}^{n \wedge d} \varphi(\widetilde{\sigma}_i) \, \widetilde{\boldsymbol{\xi}}_i \widetilde{\boldsymbol{\zeta}}_i^\top. \tag{4}$$

Given a loss function $L_n : \mathbb{R}^{n \times d} \times \mathbb{R}^{n \times d} \to \mathbb{R}_+$ quantifying the discrepancy between $\hat{S}_\varphi$ and $S$ (common choices include the Frobenius norm $\|\hat{S}_\varphi - S\|_F^2/(nd)$, operator norm, and nuclear norm), the *optimal shrinker* is defined as

$$\varphi^* := \arg\min_\varphi \lim_{n \to \infty} L_n(\hat{S}_\varphi, S), \tag{5}$$

where the limit is taken in the *high-dimensional regime* $n/d \to \beta \in (0, \infty)$ (11).

In this regime, the observed singular values $\widetilde{\sigma}_i$ are biased relative to the true signal strengths $d_i$: noise inflates the observed values, and weak signals become entangled with the noise bulk. The *Marchenko–Pastur (MP) law* (26) describes the limiting spectral distribution of $ZZ^\top$ when $Z \in \mathbb{R}^{n \times d}$ has i.i.d. entries with mean zero, variance $\sigma^2/n$, and finite fourth moment: the eigenvalues concentrate in a bulk supported on $[\lambda_-, \lambda_+]$, where $\lambda_\pm = \sigma^2(1 \pm \sqrt{\beta})^2$. A *phase transition* (4) occurs at a critical signal strength $\alpha$: signals with $d_i > \alpha$ produce outlier singular values that separate from the bulk and are detectable, while signals with $d_i \leq \alpha$ are undetectable—their singular values "stick" to the bulk edge $\sqrt{\lambda_+}$. The optimal shrinker depends on three quantities for each detectable signal $(d_i > \alpha)$: the signal strength $d_i$ and the *asymptotic singular vector overlaps*

$$a_{1,i} := \lim_{n \to \infty} |\langle \mathbf{u}_i, \tilde{\boldsymbol{\xi}}_i \rangle|^2, \qquad a_{2,i} := \lim_{n \to \infty} |\langle \mathbf{v}_i, \tilde{\boldsymbol{\zeta}}_i \rangle|^2, \tag{6}$$

which quantify how well the noisy singular vectors $\tilde{\boldsymbol{\xi}}_i$, $\tilde{\boldsymbol{\zeta}}_i$ approximate the true signal directions $\mathbf{u}_i$, $\mathbf{v}_i$. For the three standard loss functions (10; 18):

$$\varphi_i^* = \begin{cases} d_i\sqrt{a_{1,i}a_{2,i}} & \text{(Frobenius norm)}, \\ d_i\sqrt{\frac{a_{1,i} \wedge a_{2,i}}{a_{1,i} \vee a_{2,i}}} & \text{(operator norm)}, \\ d_i\left(\sqrt{a_{1,i}a_{2,i}} - \sqrt{(1-a_{1,i})(1-a_{2,i})}\right) & \text{(nuclear norm)}, \end{cases} \tag{7}$$

and $\varphi_i^* = 0$ when $d_i \leq \alpha$. Hard thresholding (truncated SVD) is a special case that retains the inflated $\widetilde{\sigma}_i$ without correction—suboptimal because the retained values are biased upward by noise and the rank must be chosen manually.

Since the high-dimensional regime requires $n$ and $d$ to be comparable $(n/d \to \beta)$, we partition the KV cache into blocks of $n$ tokens with $n$ comparable to $d$, and apply shrinkage independently to each block. This block-wise processing is natural for the streaming KV cache: as tokens arrive in chunks during prefill, each chunk forms a block that is compressed before the next chunk is processed.

When the noise has a separable covariance structure $Z = A^{1/2}XB^{1/2}$ (as in KV cache blocks, where $A$ captures temporal dependence and $B$ captures coordinate covariance), the MP law no longer applies and the bulk edge $\lambda_+$ depends on both $A$ and $B$. Standard optimal shrinkage methods (10; 11) assume white noise $(A = B = I)$ and fail in this setting. The eOptShrink estimator (33) extends the OptShrink framework (28) to handle colored noise with separable covariance, non-square matrices, and automatic rank determination—all essential for KV cache blocks where $n \neq d$ and the residual statistics are non-white. The algorithm (Algorithm 2) is fully data-driven. It estimates the bulk edge $\hat{\lambda}_+$ and effective rank $\hat{r}^+$ from the observed spectrum, recovers the noise spectral distribution $\hat{F}_e$ by imputing the eigenvalues perturbed by signals, and computes the data-driven estimates of $d_i$, $a_{1,i}$, $a_{2,i}$. Central to the algorithm is the *D-transform* (5):

$$\mathcal{T}(z) := z\, m_{1c}(z)\, m_{2c}(z), \tag{8}$$

where $m_{1c}(z)$ and $m_{2c}(z)$ are the Stieltjes transforms of the asymptotic limiting spectral measures of $ZZ^\top$ and $Z^\top Z$ respectively. The Stieltjes transform of a measure $\mu$ is $m_\mu(z) := \int (x - z)^{-1}d\mu(x)$ for $z \in \mathbb{C}^+$; see (33, Section 2) for the precise definitions of $m_{1c}$, $m_{2c}$ in terms of the noise covariance matrices $A$ and $B$. Since neither $A$, $B$, nor the noise spectral distribution are known, eOptShrink estimates them from the data. The estimated noise spectral distribution $\hat{F}_e$ is the empirical CDF of the estimated noise eigenvalues:

$$\hat{F}_e(x) = \frac{1}{n \wedge d} \sum_{i=1}^{n \wedge d} \mathbf{1}\{\hat{\lambda}_{e,i} \leq x\}, \tag{9}$$

where $\hat{\lambda}_{e,i} = \tilde{\lambda}_i$ for $i > \hat{r}^+ + k$ (non-outlier, non-perturbed eigenvalues used directly), and the top $\hat{r}^+ + k$ eigenvalues are replaced by imputed values: the $\hat{r}^+$ outlier eigenvalues are discarded, and the next $k$ eigenvalues (perturbed by proximity to the outliers) are imputed using the square-root edge behavior of the bulk (Step 2 of Algorithm 2). The estimated Stieltjes transforms $\hat{m}_{e,1,i}$, $\hat{m}_{e,2,i}$ and D-transform $\widehat{\mathcal{T}}_{e,i} := \tilde{\lambda}_i\,\hat{m}_{e,1,i}\,\hat{m}_{e,2,i}$ are then evaluated from $\hat{F}_e$. For each $1 \leq i \leq \hat{r}^+$, the estimators are (33):

$$\hat{d}_{e,i} = \frac{1}{\sqrt{\widehat{\mathcal{T}}_{e,i}}}, \quad \hat{a}_{e,1,i} = \frac{\hat{m}_{e,1,i}}{\hat{d}_{e,i}^2\,\widehat{\mathcal{T}}_{e,i}'}, \quad \hat{a}_{e,2,i} = \frac{\hat{m}_{e,2,i}}{\hat{d}_{e,i}^2\,\widehat{\mathcal{T}}_{e,i}'}, \tag{10}$$

where $\widehat{\mathcal{T}}'_{e,i}$ is the derivative of $\widehat{\mathcal{T}}_e$ at $\tilde{\lambda}_i$. These are plugged into (7) to obtain the shrunken values $\hat{\varphi}_{e,i}$, yielding $\hat{S} = \sum_{i=1}^{\hat{r}^+} \hat{\varphi}_{e,i} \, \tilde{\boldsymbol{\xi}}_i \tilde{\boldsymbol{\zeta}}_i^\top$. Crucially, the algorithm requires no knowledge of $A$, $B$, or the rank $r$—all are estimated from the data.

---

**Algorithm 2** eOptShrink (adapted from (33))

---

**Require:** Data matrix $\widetilde{S} \in \mathbb{R}^{n \times d}$ with SVD $\widetilde{S} = \sum_{i=1}^{n \wedge d} \tilde{\sigma}_i \tilde{\boldsymbol{\xi}}_i \tilde{\boldsymbol{\zeta}}_i^\top$, constant $c = \min(1/2.01, 1/\log\log d)$, loss function $L$.
1: **Output:** $\hat{S}, \hat{r}^+ \leftarrow$ eOptShrink($\widetilde{S}$)
2: **Step 1: Estimate effective rank.**
3: Set $k = \lfloor d^c \rfloor$. Estimate the bulk edge:

$$\hat{\lambda}_+ = \tilde{\lambda}_{k+1}^2 + \frac{1}{2^{2/3} - 1} \left( \tilde{\lambda}_{k+1}^2 - \tilde{\lambda}_{2k+1}^2 \right).$$

4: Set $\hat{r}^+ = |\{i : \tilde{\lambda}_i^2/\hat{\lambda}_+ - 1 > d^{-1/3}\}|$.
5: **Step 2: Estimate noise spectral distribution.**
6: Impute the top $k$ noise eigenvalues using the square-root edge behavior:

$$\hat{\lambda}_{j+\hat{r}^+}^2 = \tilde{\lambda}_{k+\hat{r}^++1}^2 + \frac{1 - (j/k)^{2/3}}{2^{2/3} - 1} \left( \tilde{\lambda}_{k+\hat{r}^++1}^2 - \tilde{\lambda}_{2k+\hat{r}^++1}^2 \right)$$

for $j = 1, \ldots, k$. Construct the estimated noise CDF $\hat{F}_e(x)$.
7: **Step 3: Compute optimal shrinker.**
8: For $1 \leq i \leq \hat{r}^+$: estimate signal strength $\hat{d}_i$ and overlaps $\hat{a}_{1,i}$, $\hat{a}_{2,i}$ via the $D$-transform of $\hat{F}_e$.
9: Compute the shrunken value $\hat{\varphi}_{e,i}$ for the chosen loss $L$ (e.g., $\hat{\varphi}_{e,i} = \hat{d}_{e,i} \sqrt{\hat{a}_{e,1,i} \hat{a}_{e,2,i}}$ for Frobenius).
10: **Return:** $\hat{S} = \sum_{i=1}^{\hat{r}^+} \hat{\varphi}_{e,i} \, \tilde{\boldsymbol{\xi}}_i \tilde{\boldsymbol{\zeta}}_i^\top, \, \hat{r}^+$.

---

# 3 Method

The eOptShrinkQ pipeline processes the KV cache in blocks of $n$ tokens (Algorithm 3). Each block $\widetilde{S} \in \mathbb{R}^{n \times d}$ is first denoised via eOptShrink (Algorithm 2), which automatically determines the rank $\hat{r}^+$ and computes the Frobenius-norm-optimal shrunken singular values. If no eigenvalue exceeds the bulk edge ($\hat{r}^+ = 0$), the block is passed directly to $\text{TQ}_{\text{MSE}}$ without SVD preprocessing. Otherwise, the low-rank estimate $\hat{S}$ is subtracted, and the residual $R = \widetilde{S} - \hat{S}$ is quantized row-by-row via $\text{TQ}_{\text{MSE}}$ (Algorithm 1). Optionally, QJL correction can be added for keys (upgrading to $\text{TQ}_{\text{prod}}$), and SVD factors can be quantized at $b_s$ bits to reduce overhead.

---

**Algorithm 3** eOptShrinkQ: Structured KV Cache Compression

---

**Require:** Data block $\widetilde{S} \in \mathbb{R}^{n \times d}$ (keys or values), residual bit-width $b$, SVD factor bit-width $b_s$

1: $\hat{S}, \hat{r}^+ \leftarrow \text{eOptShrink}(\widetilde{S})$ {Algorithm 2}
2: **if** $\hat{r}^+ > 0$ **then**
3:     (Optional) Quantize SVD factors of $\hat{S}$ at $b_s$ bits via adaptive Lloyd-Max
4:     $R \leftarrow \widetilde{S} - \hat{S}$ {isotropic residual}
5: **else**
6:     $R \leftarrow \widetilde{S}$ {no signal; skip SVD}
7: **end if**
8: **for** each row $r_t$ of $R$ **do**
9:     $\hat{r}_t \leftarrow \text{TQ}_{\text{MSE}}(r_t, b)$ {Algorithm 1}
10: **end for**
11: (Optional) For keys: apply QJL correction to upgrade $\text{TQ}_{\text{MSE}}$ to $\text{TQ}_{\text{prod}}$ (Eq. 3), adding 1 bit per entry
12: Store: shrunken SVD factors of $\hat{S}$ ($\hat{r}^+$ components), TQ indices and norms for $R$, (optional) QJL signs
13: **Bits per entry:** $b + \hat{r}^+(n+d)b_s/(nd)$ (eOptShrinkQ$_{\text{MSE}}$);   $b + 1 + \hat{r}^+(n+d)b_s/(nd)$ (eOptShrinkQ$_{\text{prod}}$)

---

[Automatic rank selection] Rather than using a fixed rank, eOptShrink determines the rank and shrunken singular values automatically from the data via the BBP phase transition threshold, without requiring knowledge of the noise covariance. When no eigenvalue exceeds the bulk edge ($\hat{r}^+ = 0$), the block is entirely isotropic and passed directly to $\text{TQ}_{\text{MSE}}$—avoiding unnecessary SVD overhead on blocks without detectable low-rank structure. In all experiments, we use the Frobenius norm loss and set $c = \min(1/2.01, 1/\log(\log d))$; for $d = 128$ this gives pilot parameter $k = 11$. SVD factors are quantized at 4 bits via data-adaptive Lloyd-Max codebooks. The bit budget formula in Algorithm 3 accounts for the residual quantization ($b$ bits per entry) and the SVD factor overhead ($\hat{r}^+(n+d)b_s/(nd)$). Two additional small costs are shared across all TQ-based methods and cancel in comparisons: (1) TQ stores one FP16 norm per row ($16/d$ bits per entry, i.e., 0.125 for $d = 128$), and (2) the shrunken singular values $\hat{\Sigma}$ are stored at FP16 ($16\hat{r}^+/(nd)$ bits per entry, negligible since $\hat{r}^+ \ll \min(n, d)$).

## 4    Numerical Experiments

We conduct numerical experiments on two model families to validate the compression pipeline: Llama-3.1-8B-Instruct (13) ($d = 128$, 32 layers × 8 KV heads) and Ministral-8B-Instruct (23) ($d = 128$, 36 layers × 8 KV heads). We first evaluate per-head reconstruction quality (Section 4.1), then end-to-end performance on LongBench and multi-needle retrieval (Section 4.2).

**Methods compared.**    We compare the following approaches:

- **KIVI** (24): per-channel asymmetric quantization for keys (axis=0) and per-token for values (axis=1), with group size 64. This is the most widely adopted KV cache quantization baseline. Total bits include the group-wise scale and zero-point overhead ($32/g$ bits per entry for group size $g$).

- **TQ$_{\text{MSE}}$**: standalone TurboQuant (40) with MSE-optimal Lloyd-Max quantization (Section 2.2).

- **TQ$_{\text{prod}}$**: TurboQuant with $b$-bit MSE plus 1-bit QJL correction for unbiased inner products ($b + 1$ total bits).

- **SVD$_{r=1}$ + TQ$_{\text{MSE}}$**: rank-1 truncated SVD to remove the leading singular component, then $\text{TQ}_{\text{MSE}}$ on the residual. This baseline isolates the effect of removing even a single dominant direction, at negligible bit overhead (0.04 bits), and demonstrates the value of spectral denoising in its simplest form.

- **eOptShrinkQ$_{\text{MSE}}$**: optimal shrinkage with data-adaptive rank (Section 3), then TQ$_{\text{MSE}}$ on the residual. Optionally, **eOptShrinkQ$_{\text{prod}}$** replaces TQ$_{\text{MSE}}$ with TQ$_{\text{prod}}$ on the residual for additional QJL correction.

SVD factors are quantized at 4 bits in all experiments. All blocks use sequential (causal) token ordering.

**No sub-grouping or outlier handling.** Unlike the original TurboQuant implementation (40) and other quantization approaches that partition coordinates into sub-vectors of different sizes and handle outlier channels separately, we apply TQ$_{\text{MSE}}$ to the full $d$-dimensional vector without sub-grouping. After eOptShrink removes the low-rank signal, the residual rows satisfy the delocalization bound $\|r_t\|_\infty \leq C\sqrt{\log d/d} \cdot \|r_t\|_2$ (Corollary 6): no coordinate is an outlier, so sub-grouping is unnecessary. For fair comparison, all methods—including the TQ$_{\text{MSE}}$ and TQ$_{\text{prod}}$ baselines—use the same full-vector quantization without sub-grouping.

## 4.1 KV Cache Compression Quality

We run Llama-3.1-8B-Instruct (13) and Ministral-8B-Instruct (23) on a LongBench (3) QAsper passage ($\sim$3000 tokens) and extract the full KV cache after prefill. We evaluate compression across all layers and KV heads (Llama: 32 layers $\times$ 8 heads = 256 heads; Ministral: 36 layers $\times$ 8 heads = 288 heads), with $d = 128$ and blocks of $128 \times 128$.

**Spiked model validation.** We first verify that the spiked model $\widetilde{S} = S + Z$ accurately describes the spectral structure of KV cache blocks. In the spiked model, the *effective rank* $r^+$ is the number of signal singular values strong enough to separate from the noise bulk (i.e., exceeding the BBP phase transition threshold $\alpha$; see Appendix A.2). eOptShrink estimates this as $\widehat{r}^+$—the number of singular values exceeding the estimated bulk edge $\sqrt{\widehat{\lambda}_+}$—without requiring knowledge of the noise distribution (Algorithm 2). Figures 1 and 5 show the singular value spectrum and distribution for representative key heads on Llama (L15H3) and Ministral (L5H3). In both cases, a small number of outlier singular values clearly separate above the estimated bulk edge $\sqrt{\widehat{\lambda}_+}$, with the remaining singular values forming a smooth bulk distribution. The leading singular value is typically 4–8$\times$ larger than the bulk edge, reflecting strong low-rank structure from local context. The clean separation validates both the spiked model assumption and the automatic rank estimation via the BBP phase transition.

Figures 3 and 7 show the corresponding spectra for value heads (Llama L31H0, Ministral L18H1). The spiked structure is also present, with outlier singular values clearly separating from the bulk, though the spectral gap is weaker than for keys—consistent with values lacking positional encoding. The bulk edge estimator $\sqrt{\widehat{\lambda}_+}$ correctly delineates the boundary for both keys and values on both models, confirming that eOptShrink's automatic rank estimation generalizes across model families.

**Spectral denoising visualization.** Figures 2–8 visualize the effect of eOptShrink on the data matrices. Each figure shows the original block $\widetilde{S}$, the extracted low-rank signal $\hat{S}$, and the residual $R = \widetilde{S} - \hat{S}$. For keys (Figures 2, 6), the original block exhibits prominent vertical stripes—per-channel outliers that KIVI's per-channel quantization is designed to handle. The rank-$\hat{r}^+$ signal $\hat{S}$ captures precisely these stripes, and the residual $R$ is visually isotropic with no discernible structure. For values (Figures 4, 8), the original block shows weaker but still visible column and row patterns; $\hat{S}$ captures these and $R$ is again structure-free. This confirms that eOptShrink removes the anisotropy at its source—the low-rank shared context—rather than engineering around its symptoms.

We report relative $L_2$ error $\|\hat{X} - X\|_F/\|X\|_F$ (in percent), inner product (IP) bias, and IP standard deviation on unit-normalized vectors (cosine similarity error). These per-head metrics isolate quantization fidelity from confounding factors in downstream evaluation (prompt formatting, generation strategy, task-specific scoring) and enable direct comparison across methods with different quantization strategies. Tables 1 and 2 show the results for Llama; Tables 3 and 4 for Ministral.

Tables 1–4 demonstrate four findings consistent across both models and both keys and values.

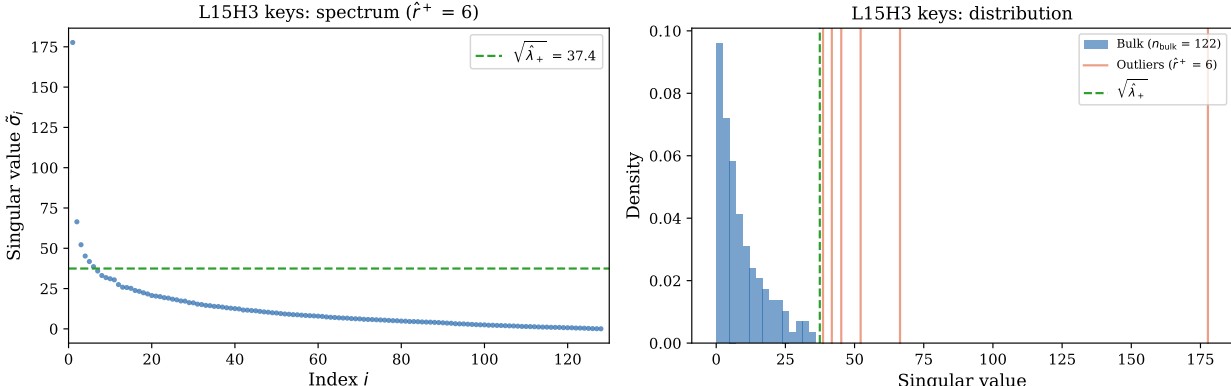

Figure 1: Spiked model validation on Llama-3.1-8B **key** vectors (L15H3, $n = 128$, $d = 128$, $\beta = 1.0$). (Left) Singular value spectrum with outliers above the estimated bulk edge $\sqrt{\widehat{\lambda}_+}$ (green dashed). (Right) Distribution of all singular values with outliers marked (coral lines).

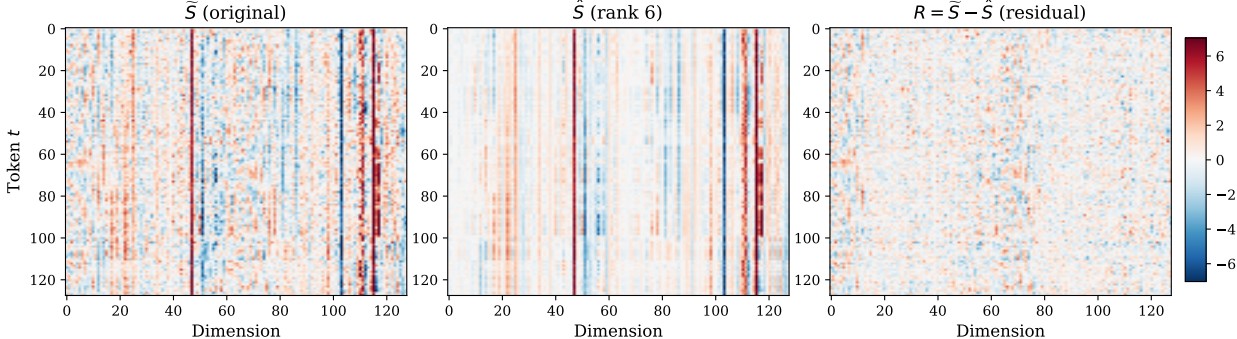

Figure 2: Spectral denoising of Llama-3.1-8B **key** block (L15H3, $128 \times 128$). (Left) Original block $\widetilde{S}$ with visible vertical stripes from per-channel outliers. (Center) The rank-$\hat{r}^+$ signal $\hat{S}$ extracted by eOptShrink. (Right) The residual $R = \widetilde{S} - \hat{S}$ with no visible structure—the isotropic matrix that $\mathrm{TQ}_{\mathrm{MSE}}$ quantizes.

*eOptShrinkQ outperforms KIVI at fewer bits.* KIVI (24) uses per-channel quantization for keys and per-token quantization for values, with group-wise scales adding 0.5 bits overhead (group size 64). On keys, eOptShrinkQ$_{\mathrm{MSE}}$ at 2.35 bits achieves 17.7% $L_2$ versus KIVI's 24.8% at 2.50 bits—a 29% relative reduction with fewer bits. On values, KIVI's per-token quantization is particularly ineffective at low bit rates: 49.5% $L_2$ at 2.50 bits versus $\mathrm{TQ}_{\mathrm{MSE}}$'s 34.1% at 2.00 bits, because per-token quantization does not exploit the random rotation that TurboQuant uses to isotropize each vector before scalar quantization. KIVI's asymmetric treatment of keys versus values (different quantization axes) is an engineering response to the anisotropy caused by the low-rank structure; our approach removes the root cause, enabling uniform treatment with a single quantizer. We note that KIVI's advantage lies in hardware efficiency: its per-channel INT quantization maps directly to tensor core operations, whereas TurboQuant's Lloyd-Max codebooks require lookup tables. Our contribution is orthogonal—spectral denoising can in principle be combined with any downstream quantizer, including hardware-friendly INT schemes.

*eOptShrinkQ saves nearly one bit per entry over TurboQuant.* At every bit rate, eOptShrinkQ$_{\mathrm{MSE}}$ achieves the lowest $L_2$ error and IP standard deviation. Notably, eOptShrinkQ$_{\mathrm{MSE}}$ at $b = 2$ (2.35 total bits) achieves 17.7% $L_2$ on Llama keys—substantially better than KIVI at 2.50 bits (24.8%) and $\mathrm{TQ}_{\mathrm{MSE}}$ at 2.00 bits (34.1%). At $b = 3$, eOptShrinkQ$_{\mathrm{MSE}}$ at 3.35 bits (9.6% $L_2$) outperforms $\mathrm{TQ}_{\mathrm{MSE}}$ at 4.00 bits (9.7%)—achieving better reconstruction with 0.65 fewer bits. This bit saving comes from the spectral denoising: by extracting the low-rank structure ($\bar{r} \approx 5$–6), the residual becomes more isotropic and $\mathrm{TQ}_{\mathrm{MSE}}$ operates

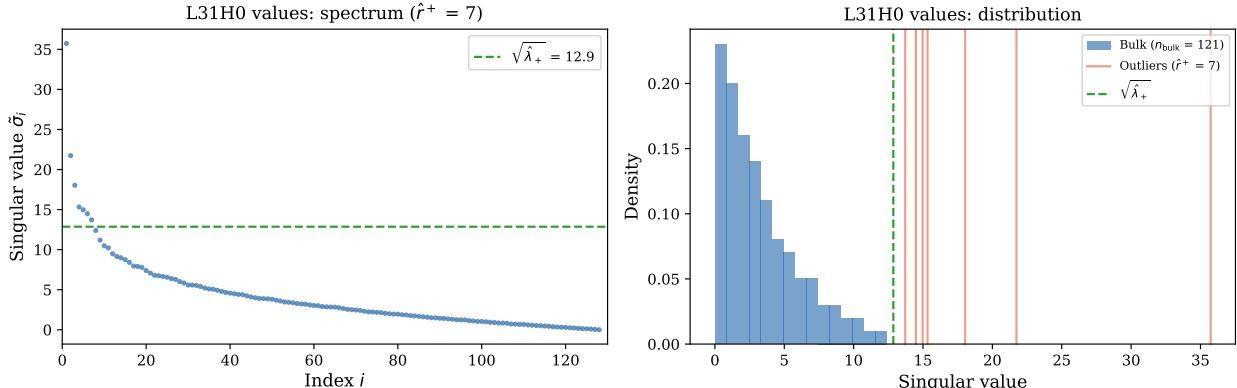

Figure 3: Spiked model validation on Llama-3.1-8B **value** vectors (L31H0, $n = 128$, $d = 128$, $\beta = 1.0$). Values exhibit weaker but still detectable low-rank structure, with outliers closer to the bulk edge than keys.

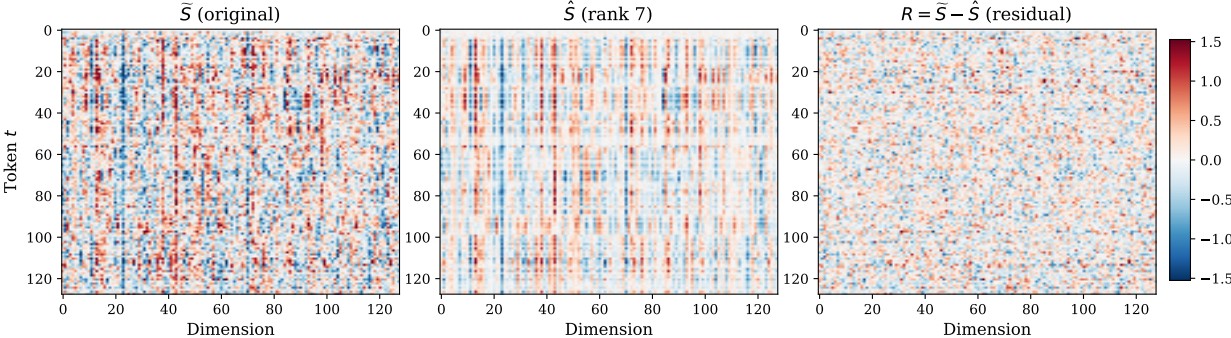

Figure 4: Spectral denoising of Llama-3.1-8B **value** block (L31H0, $128 \times 128$). Values show weaker but still visible block structure in $\widetilde{S}$, captured by $\hat{S}$ and absent from the residual $R$.

closer to its theoretical optimum. The SVD overhead is $\hat{r}^+(n+d)b_s/(nd)$ bits per entry (Algorithm 3): for $\text{SVD}_{r=1}$ with $n = 128$, $d = 128$, $b_s = 4$, this is $1 \times 256 \times 4/16384 \approx 0.06$ bits; for eOptShrinkQ with $\bar{r} \approx 5.6$, approximately 0.35 bits.

*QJL correction is largely unnecessary after spectral denoising.* $\text{TQ}_{\text{prod}}$ eliminates IP bias but at a cost: it *increases* IP standard deviation compared to $\text{TQ}_{\text{MSE}}$ at the same MSE bit rate (e.g., .036 vs. .026 at $b = 2$ on Llama keys), because the QJL correction trades bias for variance. $\text{eOptShrinkQ}_{\text{MSE}}$ achieves near-zero bias ($+.006$ at $b = 2$, vanishing at $b = 4$) *and* the lowest IP standard deviation simultaneously—without spending any bits on QJL. Adding QJL on top ($\text{eOptShrinkQ}_{\text{prod}}$) drives the bias to exactly zero but increases the variance, making it a marginal improvement at best.

*Automatic rank selection is essential.* $\text{SVD}_{r=1}$ truncation already provides a large improvement over $\text{TQ}_{\text{MSE}}$ (e.g., 21.2% vs. 34.1% $L_2$ at $b = 2$ on Llama keys), demonstrating the value of removing even a single dominant component. However, eOptShrinkQ with adaptive rank $\bar{r} \approx 5$–6 reduces the error further to 17.8%—a 16% relative improvement over $\text{SVD}_{r=1}$. The gap is even larger on values, where the spectral structure is distributed across more components: on Llama values at $b = 2$, eOptShrinkQ achieves 26.4% vs. $\text{SVD}_{r=1}$'s 30.4%. The higher rank extracts more shared structure, leaving a more isotropic residual that $\text{TQ}_{\text{MSE}}$ compresses more efficiently.

## 4.2 End-to-End Evaluation

We evaluate end-to-end on LongBench (3) (16 English tasks) and multi-needle retrieval using Llama-3.1-8B-Instruct (13) and Ministral-8B-Instruct (23). Compression is applied chunk-by-chunk during prefill: each

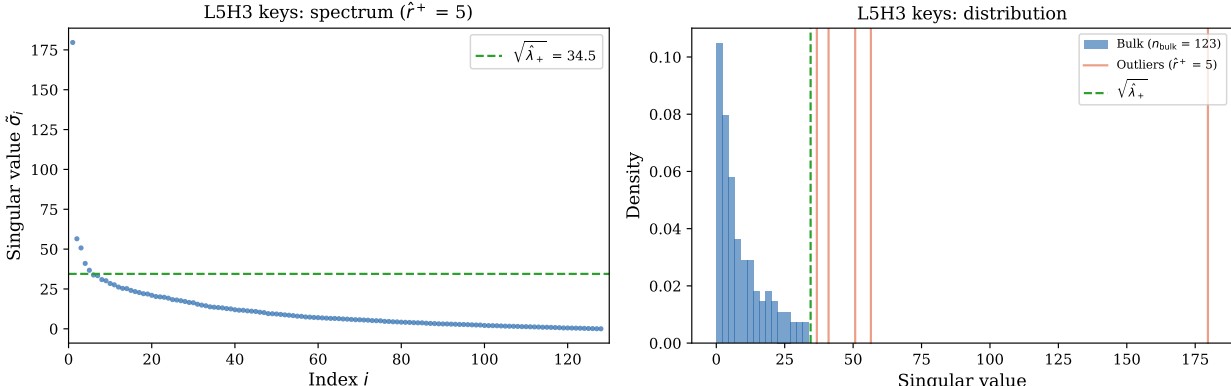

Figure 5: Spiked model validation on Ministral-8B **key** vectors (L5H3, $n = 128$, $d = 128$, $\beta = 1.0$). Same format as Figure 1. The spiked structure is consistent across model families.

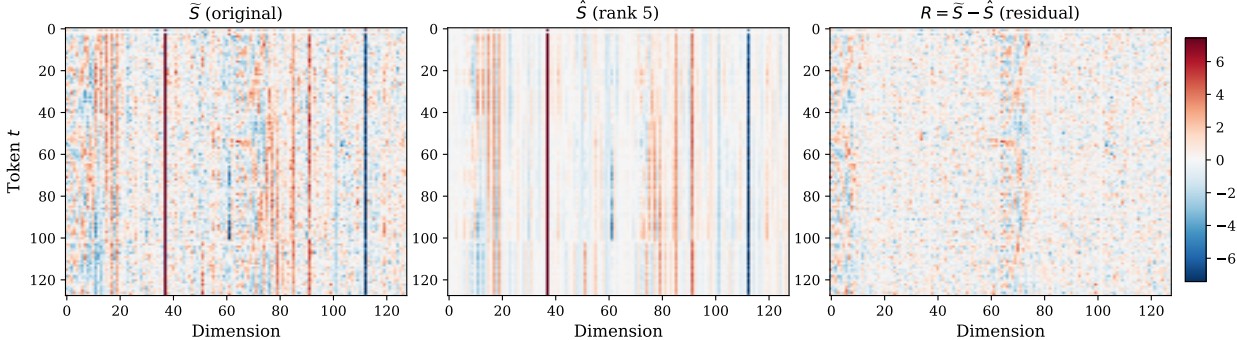

Figure 6: Spectral denoising of Ministral-8B **key** block (L5H3, $128 \times 128$). Same format as Figure 2. The vertical stripe structure and its removal by eOptShrink are consistent with Llama.

chunk of 128 tokens passes through all layers in FP16, then the KV cache for that chunk is compressed in-place using blocks of $128 \times 128$. Both keys and values are compressed at the same bit rate $b$. This captures two levels of error accumulation: *between layers*, where layer $l + 1$ computes its hidden states from layer $l$'s compressed attention output, and *between chunks*, where subsequent chunks attend over already-compressed cache from all previous chunks. For a sequence of $T$ tokens with $L$ layers, compression errors propagate through $\lfloor T/128 \rfloor \times L$ compression-then-attend steps.

**LongBench.** Table 5 shows category-averaged scores across all 16 tasks. eOptShrinkQ$_{\text{MSE}}$ at ~2.2 bits achieves 47.4 on Llama and 48.3 on Ministral—within 1.6 and 2.2 points of FP16 respectively, and outperforming TQ$_{\text{prod}}$ at 3.0 bits (45.2 and 46.6) despite using substantially fewer bits. eOptShrinkQ$_{\text{MSE}}$ wins 4 of 6 categories on both models, with especially large gains on MultiQA (+4.8 over TQ$_{\text{MSE}}$ on Llama) and Code (+4.7), where precise retrieval of specific tokens across long contexts is essential. On Code, eOptShrinkQ$_{\text{MSE}}$ at 2.2 bits (53.9) matches FP16 (53.1), suggesting that spectral regularization can benefit retrieval-intensive tasks. Adding QJL (eOptShrinkQ$_{\text{prod}}$ at ~3.2 bits) provides a small further gain to 47.6, primarily on Summarization and SingleQA.

**Multi-Needle Retrieval.** Needle-in-a-Haystack (30) tests whether a model can retrieve a planted fact from a long context. We extend this to the multi-needle setting: $N$ distinct facts are scattered at evenly spaced positions throughout the context, and the model must retrieve all $N$ simultaneously. Figures 9 and 10 show recall heatmaps across $N = 2, 4, 6, 8, 10$ needles and context lengths from 4K to 104K tokens (Llama) and 4K to 30K (Ministral). eOptShrinkQ$_{\text{MSE}}$ at 2.2 bits achieves 0.981 average recall on Llama— outperforming FP16 (0.972)—and 0.992 on Ministral, nearly matching FP16 (1.000). The heatmaps reveal

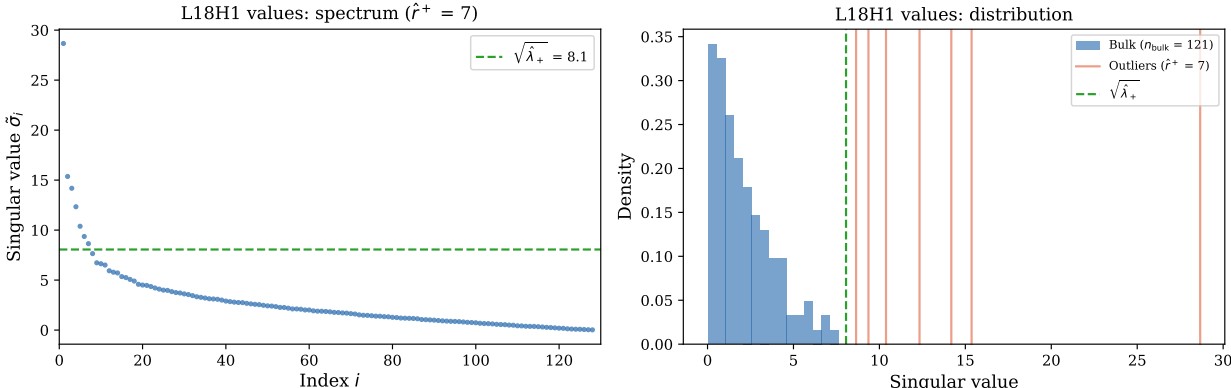

Figure 7: Spiked model validation on Ministral-8B **value** vectors (L18H1, $n = 128$, $d = 128$, $\beta = 1.0$). Same format as Figure 3.

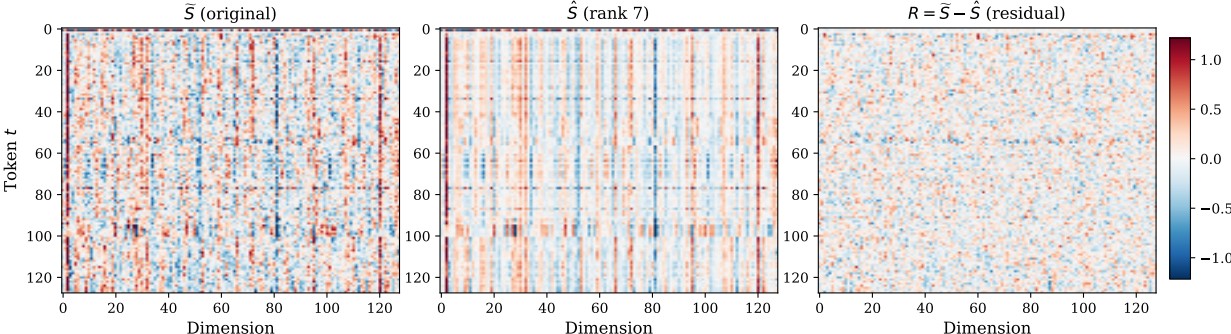

Figure 8: Spectral denoising of Ministral-8B **value** block (L18H1, $128 \times 128$). Same format as Figure 4.

that $\mathrm{SVD}_{r=1}$ produces scattered failures across lengths and needle counts (average 0.886 on Llama), while $\mathrm{eOptShrinkQ_{MSE}}$ maintains near-perfect recall throughout. $\mathrm{TQ_{prod}}$ at 3.0 bits is the worst method on both models (0.940 and 0.938), as the QJL correction increases inner product variance that accumulates across the many compressed tokens the model must attend over during multi-fact retrieval.

## 5 Analysis

We highlight three observations from the experiments that connect to the theoretical framework.

*Spectral regularization.* On tasks requiring precise token-level retrieval, $\mathrm{eOptShrinkQ_{MSE}}$ matches or exceeds FP16: Code on Llama (53.9 vs. 53.1) and multi-needle retrieval on Llama (0.981 vs. 0.972). This effect is task-dependent: on Summarization, which requires integrating shared context across the entire passage, $\mathrm{eOptShrinkQ_{MSE}}$ slightly underperforms FP16 (27.4 vs. 29.2 on Llama). The pattern suggests that removing the low-rank shared context acts as a form of spectral regularization: it reduces the effective dimension of the attention computation, benefiting tasks where attention must discriminate among many similar token representations (retrieval, code) while slightly degrading tasks that rely on global context integration (summarization). Notably, $\mathrm{eOptShrinkQ_{prod}}$ at 3.22 bits partially recovers the summarization gap (28.5 vs. 29.2), suggesting that the QJL correction specifically helps tasks where preserving the shared structure's contribution to inner products matters. On Ministral, $\mathrm{eOptShrinkQ_{MSE}}$ closely tracks FP16 across all categories without exceeding it, suggesting the regularization effect is model- and task-dependent rather than a universal property.

Table 1: Compression quality on Llama-3.1-8B-Instruct **keys** ($d = 128$, blocks of $128 \times 128$, averaged over 256 heads). SVD factors quantized at 4 bits. $\bar{r}$: mean eOptShrink rank. IP reported as bias $\pm$ std on unit-normalized vectors.

| Method | $\bar{r}$ | $b = 2$ | | | $b = 3$ | | | $b = 4$ | | |
| | | Bits | $L_2\%$ | IP | Bits | $L_2\%$ | IP | Bits | $L_2\%$ | IP |
|---|---|---|---|---|---|---|---|---|---|---|
| KIVI | — | 2.50 | 24.8 | $-.016 \pm .022$ | 3.50 | 10.5 | $-.003 \pm .010$ | 4.50 | 4.9 | $-.001 \pm .004$ |
| TQ$_{\text{MSE}}$ | — | 2.00 | 34.1 | $-.028 \pm .027$ | 3.00 | 18.5 | $-.008 \pm .014$ | 4.00 | 9.7 | $-.002 \pm .007$ |
| TQ$_{\text{prod}}$ | — | 3.00 | 34.1 | $-.001 \pm .036$ | 4.00 | 18.5 | $.000 \pm .020$ | 5.00 | 9.7 | $.000 \pm .010$ |
| SVD$_{r=1}$+TQ | 1.0 | 2.06 | 21.3 | $+.009 \pm .017$ | 3.06 | 11.5 | $+.003 \pm .010$ | 4.06 | 6.0 | $+.001 \pm .005$ |
| eOptShrinkQ$_{\text{MSE}}$ | 5.6 | 2.35 | **17.7** | $+.006 \pm \mathbf{.014}$ | 3.35 | **9.6** | $+.002 \pm \mathbf{.008}$ | 4.35 | **5.0** | $.000 \pm \mathbf{.004}$ |
| eOptShrinkQ$_{\text{prod}}$ | 5.6 | 3.35 | **17.7** | $\mathbf{.000} \pm .019$ | 4.35 | **9.6** | $\mathbf{.000} \pm .011$ | 5.35 | **5.0** | $\mathbf{.000} \pm .006$ |

Table 2: Compression quality on Llama-3.1-8B-Instruct **values** ($d = 128$, blocks of $128 \times 128$, averaged over 256 heads). Format as in Table 1.

| Method | $\bar{r}$ | $b = 2$ | | | $b = 3$ | | | $b = 4$ | | |
| | | Bits | $L_2\%$ | IP | Bits | $L_2\%$ | IP | Bits | $L_2\%$ | IP |
|---|---|---|---|---|---|---|---|---|---|---|
| KIVI | — | 2.50 | 49.5 | $-.012 \pm .041$ | 3.50 | 21.2 | $-.003 \pm .018$ | 4.50 | 9.9 | $-.001 \pm .008$ |
| TQ$_{\text{MSE}}$ | — | 2.00 | 34.1 | $-.007 \pm .031$ | 3.00 | 18.4 | $-.002 \pm .016$ | 4.00 | 9.7 | $-.001 \pm .008$ |
| TQ$_{\text{prod}}$ | — | 3.00 | 34.1 | $.000 \pm .037$ | 4.00 | 18.4 | $.000 \pm .020$ | 5.00 | 9.7 | $.000 \pm .010$ |
| SVD$_{r=1}$+TQ | 1.0 | 2.06 | 30.6 | $+.004 \pm .028$ | 3.06 | 16.5 | $+.001 \pm .015$ | 4.06 | 8.7 | $.000 \pm .008$ |
| eOptShrinkQ$_{\text{MSE}}$ | 5.0 | 2.31 | **26.6** | $+.003 \pm \mathbf{.024}$ | 3.31 | **14.4** | $+.001 \pm \mathbf{.013}$ | 4.31 | **7.5** | $.000 \pm \mathbf{.007}$ |
| eOptShrinkQ$_{\text{prod}}$ | 5.0 | 3.31 | **26.6** | $\mathbf{.000} \pm .030$ | 4.31 | **14.4** | $\mathbf{.000} \pm .016$ | 5.31 | **7.5** | $\mathbf{.000} \pm .008$ |

*QJL correction trades bias for variance.* TQ$_{\text{prod}}$ achieves near-zero IP bias but at the cost of increased IP standard deviation (e.g., .036 vs. .026 for TQ$_{\text{MSE}}$ at $b = 2$ on Llama keys, Table 1). On downstream tasks, this trade-off is unfavorable: TQ$_{\text{prod}}$ is consistently the worst method on multi-needle retrieval (0.940 on Llama, 0.938 on Ministral), because the variance from QJL correction accumulates across the many compressed tokens that the model must attend over during generation. eOptShrinkQ$_{\text{MSE}}$ achieves near-zero bias *and* the lowest variance simultaneously, without spending any bits on QJL. This explains why eOptShrinkQ$_{\text{MSE}}$ at 2.22 bits outperforms TQ$_{\text{prod}}$ at 3.0 bits across all evaluations.

*Adaptive rank across layers.* The effective rank $\hat{r}^+$ selected by eOptShrink varies across layers and heads: on Llama keys ($n = 128$, $d = 128$), we observe $\hat{r}^+ \approx 9$ in early layers, $\hat{r}^+ \approx 11$ in middle layers, and $\hat{r}^+ \approx 5$ in the final layer. This variation reflects the changing nature of contextual structure across the network: middle layers tend to carry richer semantic representations, while final layers are more diffuse—a pattern consistent with the spectral analysis of KV caches reported in CSKV (37). Fixed-rank methods like SVD$_{r=1}$ cannot capture this variation, which explains their inferior performance especially on values ($\bar{r} \approx 5$) where the low-rank structure is distributed across more components than a single direction.

## 6    Theoretical Guarantees

We now formalize the key theoretical result: after optimal shrinkage, the residual enters the regime where per-vector scalar quantization achieves near-optimal distortion with negligible inner product bias. We state the result under the spiked model $\widetilde{S} = S + Z$ with separable noise covariance $Z = A^{1/2} X B^{1/2}$ (Eq. (14)–(15)), assuming signal strengths exceed the BBP phase transition threshold and signal directions are independent of the noise. The precise technical conditions (moment bounds, aspect ratio, covariance regularity) are given in Assumption A.2 in Appendix A.2. We use the stochastic dominance notation $\xi \prec \zeta$ to denote high-probability bounds up to sub-polynomial factors (Appendix A.3).

[Residual isotropy after optimal shrinkage] Under Assumption A.2 (which treats signal singular vectors as random; see Remark A.2 for the scope of this assumption at inference time), let $\hat{S}$ be the eOptShrink estimate

Table 3: Compression quality on Ministral-8B-Instruct **keys** ($d = 128$, blocks of $128 \times 128$, averaged over 288 heads). Format as in Table 1.

| Method | $\bar{r}$ | $b = 2$ | | | $b = 3$ | | | $b = 4$ | | |
| --- | --- | --- | --- | --- | --- | --- | --- | --- | --- | --- |
| | | Bits | $L_2\%$ | IP | Bits | $L_2\%$ | IP | Bits | $L_2\%$ | IP |
| KIVI | — | 2.50 | 25.2 | $-.016 \pm .023$ | 3.50 | 10.7 | $-.003 \pm .010$ | 4.50 | 5.0 | $-.001 \pm .005$ |
| $\text{TQ}_{\text{MSE}}$ | — | 2.00 | 34.2 | $-.029 \pm .027$ | 3.00 | 18.5 | $-.008 \pm .014$ | 4.00 | 9.7 | $-.002 \pm .007$ |
| $\text{TQ}_{\text{prod}}$ | — | 3.00 | 34.2 | $-.002 \pm .036$ | 4.00 | 18.5 | $-.001 \pm .020$ | 5.00 | 9.7 | $.000 \pm .010$ |
| $\text{SVD}_{r=1}+\text{TQ}$ | 1.0 | 2.06 | 21.4 | $+.009 \pm .017$ | 3.06 | 11.6 | $+.003 \pm .010$ | 4.06 | 6.1 | $+.001 \pm .005$ |
| $\text{eOptShrinkQ}_{\text{MSE}}$ | 5.5 | 2.34 | **18.1** | $+.005 \pm \mathbf{.014}$ | 3.34 | **9.8** | $+.002 \pm \mathbf{.008}$ | 4.34 | **5.1** | $.000 \pm \mathbf{.004}$ |
| $\text{eOptShrinkQ}_{\text{prod}}$ | 5.5 | 3.34 | **18.1** | $.000 \pm .020$ | 4.34 | **9.8** | $.000 \pm .011$ | 5.34 | **5.1** | $.000 \pm .006$ |

Table 4: Compression quality on Ministral-8B-Instruct **values** ($d = 128$, blocks of $128 \times 128$, averaged over 288 heads). Format as in Table 1.

| Method | $\bar{r}$ | $b = 2$ | | | $b = 3$ | | | $b = 4$ | | |
| --- | --- | --- | --- | --- | --- | --- | --- | --- | --- | --- |
| | | Bits | $L_2\%$ | IP | Bits | $L_2\%$ | IP | Bits | $L_2\%$ | IP |
| KIVI | — | 2.50 | 48.8 | $-.009 \pm .040$ | 3.50 | 20.9 | $-.002 \pm .018$ | 4.50 | 9.7 | $.000 \pm .008$ |
| $\text{TQ}_{\text{MSE}}$ | — | 2.00 | 34.1 | $-.005 \pm .031$ | 3.00 | 18.4 | $-.002 \pm .016$ | 4.00 | 9.7 | $.000 \pm .008$ |
| $\text{TQ}_{\text{prod}}$ | — | 3.00 | 34.1 | $.000 \pm .037$ | 4.00 | 18.4 | $.000 \pm .020$ | 5.00 | 9.7 | $.000 \pm .011$ |
| $\text{SVD}_{r=1}+\text{TQ}$ | 1.0 | 2.06 | 31.0 | $+.003 \pm .028$ | 3.06 | 16.8 | $+.001 \pm .015$ | 4.06 | 8.8 | $.000 \pm .008$ |
| $\text{eOptShrinkQ}_{\text{MSE}}$ | 5.0 | 2.31 | **26.8** | $+.002 \pm \mathbf{.025}$ | 3.31 | **14.5** | $+.001 \pm \mathbf{.013}$ | 4.31 | **7.6** | $.000 \pm \mathbf{.007}$ |
| $\text{eOptShrinkQ}_{\text{prod}}$ | 5.0 | 3.31 | **26.8** | $.000 \pm .030$ | 4.31 | **14.5** | $.000 \pm .016$ | 5.31 | **7.6** | $.000 \pm .008$ |

of $S$ from (14), and let $R = \widetilde{S} - \hat{S}$ be the residual. Denote the rows of $\widetilde{S}$, $\hat{S}$, $R$, and $Z$ as $x_t$, $\hat{s}_t$, $r_t$, and $z_t$ respectively for $t = 1, \ldots, n$. Then:

1. *(Residual spectrum matches noise.)* The empirical spectral distribution of $RR^\top$ converges to that of $ZZ^\top$; in particular, $R$ has no outlier singular values.

2. *(Reduced Frobenius norm.)* The residual energy concentrates at the noise level:

$$\left| \frac{1}{nd} \|R\|_F^2 - \frac{1}{nd} \|Z\|_F^2 \right| \prec \phi_d + d^{-1/2}/\Delta_{\min}, \tag{11}$$

where $\Delta_{\min} := \min_{1 \leq i \leq r^+} \Delta(d_i)$. In particular, $\|R\|_F^2 = \|Z\|_F^2 + O_\prec(nd \cdot (\phi_d + d^{-1/2}/\Delta_{\min}))$, so $\|R\|_F < \left\| \widetilde{S} \right\|_F$ whenever the signal has non-negligible energy.

3. *(Inner product bias reduction.)* Let $q \in \mathbb{R}^d$ be a query vector, and let $\tilde{r}_t$ be the TurboQuant$_{\text{MSE}}$ reconstruction of $r_t$ at $b$ bits. The inner product bias of the shrinkage-based pipeline is reduced by a factor of $1/(1 + \text{SNR}_t)$ compared to direct quantization:

$$\frac{|\mathbb{E}[\langle q, \tilde{r}_t \rangle] - \langle q, r_t \rangle|}{|\mathbb{E}[\langle q, \tilde{x}_t \rangle] - \langle q, x_t \rangle|} \leq \frac{\|r_t\|^2}{\|x_t\|^2} \leq \frac{1}{1 + \text{SNR}_t} + o_\prec(1), \tag{12}$$

where $\text{SNR}_t := \|s_t\|^2 / \|z_t\|^2$ is the per-token signal-to-noise ratio.

Part 1 follows from the eigenvalue sticking theorem and delocalization of non-outlier singular vectors (33, Theorems 3.3–3.4); Part 2 from concentration of the cross term $\langle S - \hat{S}, Z \rangle_F$; Part 3 from applying TurboQuant's bias bound (Theorem 1 of (40)) to the residual versus the original vector. The full proof is given in Appendix A.6.

[Why QJL becomes unnecessary] The delocalization property is the theoretical reason why QJL correction becomes unnecessary after shrinkage. TurboQuant$_{\text{prod}}$'s QJL correction guarantees unbiased IP estimation

Table 5: LongBench scores by category and overall average (16 English tasks, residual bit-width $b = 2$; total bits per entry shown in the Bits column). Both keys and values compressed at the same rate. Chunk-based evaluation with 128-token chunks and $128 \times 128$ blocks. SVD factors quantized at 4 bits. Best compressed method per category in **bold**.

| Method | Bits | Llama-3.1-8B-Instruct | | | | | | |
|---|---|---|---|---|---|---|---|---|
| | | SingQA | MultQA | Summ | Few | Synth | Code | Avg |
| FP16 | 16 | 43.5 | 44.9 | 29.2 | 69.4 | 53.9 | 53.1 | 49.0 |
| TQ$_{\text{MSE}}$ | 2.00 | 41.0 | 37.9 | 27.1 | 67.8 | 46.4 | 49.2 | 44.9 |
| TQ$_{\text{prod}}$ | 3.00 | 41.9 | 38.6 | 27.6 | 67.5 | 47.2 | 48.5 | 45.2 |
| SVD$_{r=1}$+TQ | 2.04 | 41.5 | 39.4 | 26.3 | 66.9 | 49.4 | 52.2 | 46.0 |
| eOptShrinkQ$_{\text{MSE}}$ | 2.22 | 42.3 | **42.7** | 27.4 | 67.4 | **50.5** | **53.9** | 47.4 |
| eOptShrinkQ$_{\text{prod}}$ | 3.22 | **43.1** | 42.6 | **28.5** | **67.9** | 50.5 | 53.1 | **47.6** |
| Method | Bits | Ministral-8B-Instruct | | | | | | |
| | | SingQA | MultQA | Summ | Few | Synth | Code | Avg |
| FP16 | 16 | 42.5 | 49.5 | 27.8 | 71.0 | 54.5 | 57.3 | 50.5 |
| TQ$_{\text{MSE}}$ | 2.00 | 39.3 | 42.6 | 26.2 | 68.4 | 48.2 | 55.0 | 46.6 |
| TQ$_{\text{prod}}$ | 3.00 | 39.5 | 42.6 | 26.7 | 68.5 | 47.8 | 54.5 | 46.6 |
| SVD$_{r=1}$+TQ | 2.04 | 39.8 | 42.3 | 25.5 | 68.4 | 47.5 | 55.7 | 46.5 |
| eOptShrinkQ$_{\text{MSE}}$ | 2.22 | 40.1 | **46.6** | 26.3 | 69.2 | 50.5 | 56.9 | 48.3 |
| eOptShrinkQ$_{\text{prod}}$ | 3.22 | **40.9** | 46.3 | **27.1** | **69.4** | **50.8** | **57.0** | **48.6** |

for *any* input, but costs 1 bit per coordinate. After eOptShrink, the residual rows $r_t$ have small norms and delocalized spectral structure—precisely the regime where TurboQuant$_{\text{MSE}}$'s bias is already negligible. Spending that bit on MSE quality (which scales as $4^{-b}$) yields a $4\times$ improvement in distortion compared to dedicating it to QJL.

[Coordinate delocalization of residual rows] Under Assumption A.2, the rows of the residual $R = \widetilde{S} - \hat{S}$ satisfy the following coordinate delocalization: for each row $t = 1, \ldots, n$,

$$\frac{\|r_t\|_\infty}{\|r_t\|_2} \leq C \sqrt{\frac{\log d}{d}} \tag{13}$$

with high probability, where $C > 0$ is an absolute constant. In particular, the unit-normed residual rows $r_t / \|r_t\|$ are approximately uniformly distributed on the sphere $\mathcal{S}^{d-1}$ in the sense that no coordinate carries disproportionate energy, which is the precondition for TurboQuant's near-optimal quantization (40, Lemma 1).

The proof is given in Appendix A.7.

[Why shrinkage improves quantization] TQ$_{\text{MSE}}$ assumes each input vector is approximately uniform on $\mathcal{S}^{d-1}$ (40, Lemma 1). KV cache vectors violate this: within a block, unit-normed directions cluster around the dominant right singular vectors of $\widetilde{S}$, so after random rotation the coordinate marginals are no longer i.i.d. $\mathcal{N}(0, 1/d)$ and the Lloyd-Max codebook becomes suboptimal. By removing $S$ via eOptShrink, the residual rows satisfy the thin shell and delocalization properties of $Z$ (Section 1, Corollary 6), restoring the uniformity assumption. The low-rank component $S$ is stored separately via quantized SVD factors at negligible overhead.

[Inner product bias reduction] The MSE quantizer produces a negative IP bias $\mathbb{E}[\langle y, \tilde{x} \rangle] < \langle y, x \rangle$ that scales with $\|x\|^2$. Our decomposition $\langle y, x \rangle = \langle y, \hat{x} \rangle + \langle y, r \rangle$ splits this into: (1) the low-rank term $\langle y, \hat{x} \rangle$, computed from precisely stored SVD factors with negligible error, and (2) the residual term $\langle y, r \rangle$, quantized via TQ$_{\text{MSE}}$ with bias scaling as $\|r\|^2 \ll \|x\|^2$. Since the bias is reduced by a factor of $1/(1 + \text{SNR}_t)$ (Proposition 6, Part 3), all $b$ bits can be devoted to MSE reconstruction. QJL correction (TQ$_{\text{prod}}$) can optionally be added on top for further improvement.

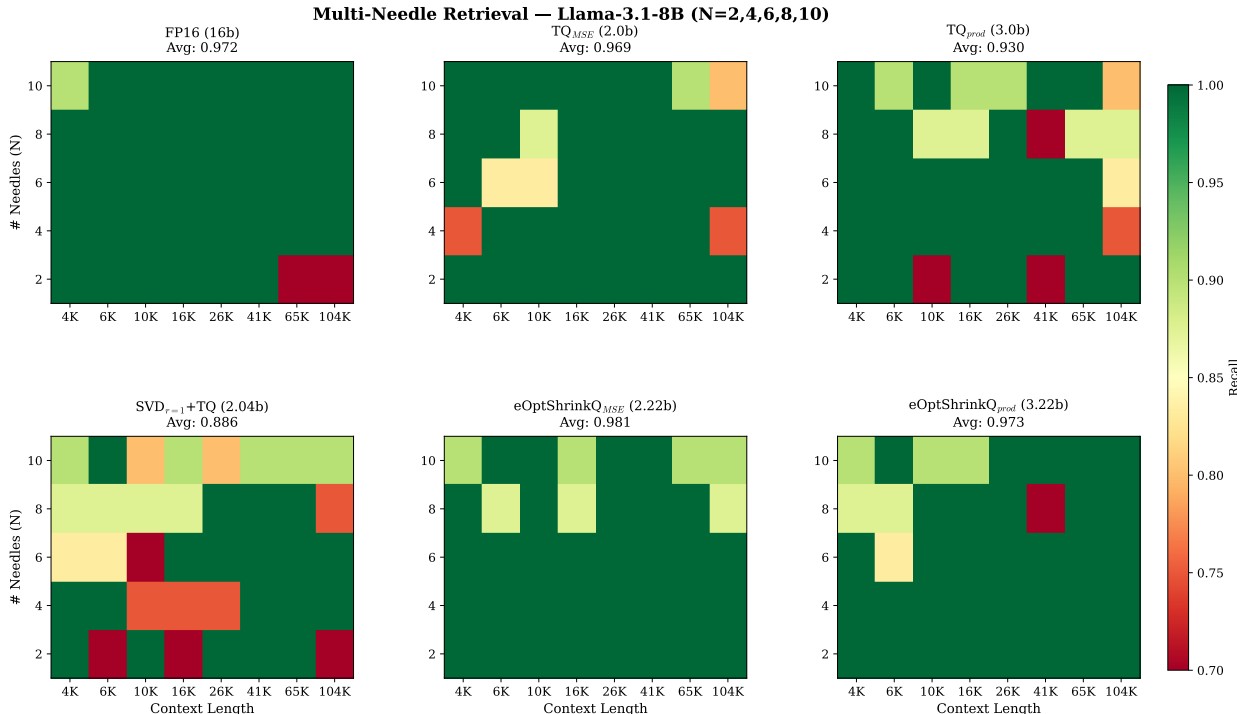

Figure 9: Multi-needle retrieval heatmaps for Llama-3.1-8B ($N = 2, 4, 6, 8, 10$ needles, context 4K–104K). Each cell shows recall (fraction of needles retrieved). eOptShrinkQ$_{\text{MSE}}$ at 2.22 bits (avg 0.981) exceeds FP16 (0.972) on this benchmark, though the effect is task- and model-dependent (see Section 5). SVD$_{r=1}$ shows scattered failures; TQ$_{\text{prod}}$ is worst despite using more bits.

# 7 Related Work

**KV Cache Quantization.** Several approaches have been proposed to reduce KV cache memory through quantization. KIVI (24) uses per-channel quantization with 2-bit keys and 5-bit values. KVQuant (15) extends this with per-channel rotation and non-uniform datatypes. PolarQuant (**?** ) applies polar transformation before quantization. GEAR (17) quantizes the bulk of KV entries to ultra-low precision, then approximates the *quantization error* with a low-rank matrix and corrects outliers with a sparse matrix— an error-correction approach that is complementary to our signal-extraction pipeline. Dynamic Memory Compression (29) merges KV cache entries via learned aggregation. SnapKV (22) and PyramidKV (6) use attention-based token eviction to reduce cache size. Our approach is complementary to eviction and merging methods: quantization reduces per-token cost while eviction and merging reduce token count.

**Low-Rank KV Cache Compression.** A growing body of work applies SVD to KV caches for *dimensionality reduction*: Palu (8) decomposes KV projection weights via SVD to cache lower-dimensional intermediate states; CSKV (37) analyzes the singular value distribution of KV caches and applies low-rank decomposition to shrink channels; SVDq (39) projects keys into an SVD basis with importance-aware quantization of latent channels; xKV (7) exploits cross-layer singular vector alignment for shared-basis compression; and OjaKV (41) uses online PCA via Oja's rule for adaptive low-rank projection.

Our approach differs fundamentally from the above. These methods use SVD for *dimensionality reduction*: they project KV vectors into a lower-dimensional subspace, store the compressed representation, and reconstruct at inference time. In contrast, we use SVD for *spectral denoising before quantization*: the residual after optimal shrinkage remains full-dimensional and is quantized by a downstream per-vector scalar quantizer. The low-rank component is a preprocessing step that restores the isotropy assumptions required by the quantizer, rather than the compression mechanism itself. Furthermore, all existing methods use truncated SVD

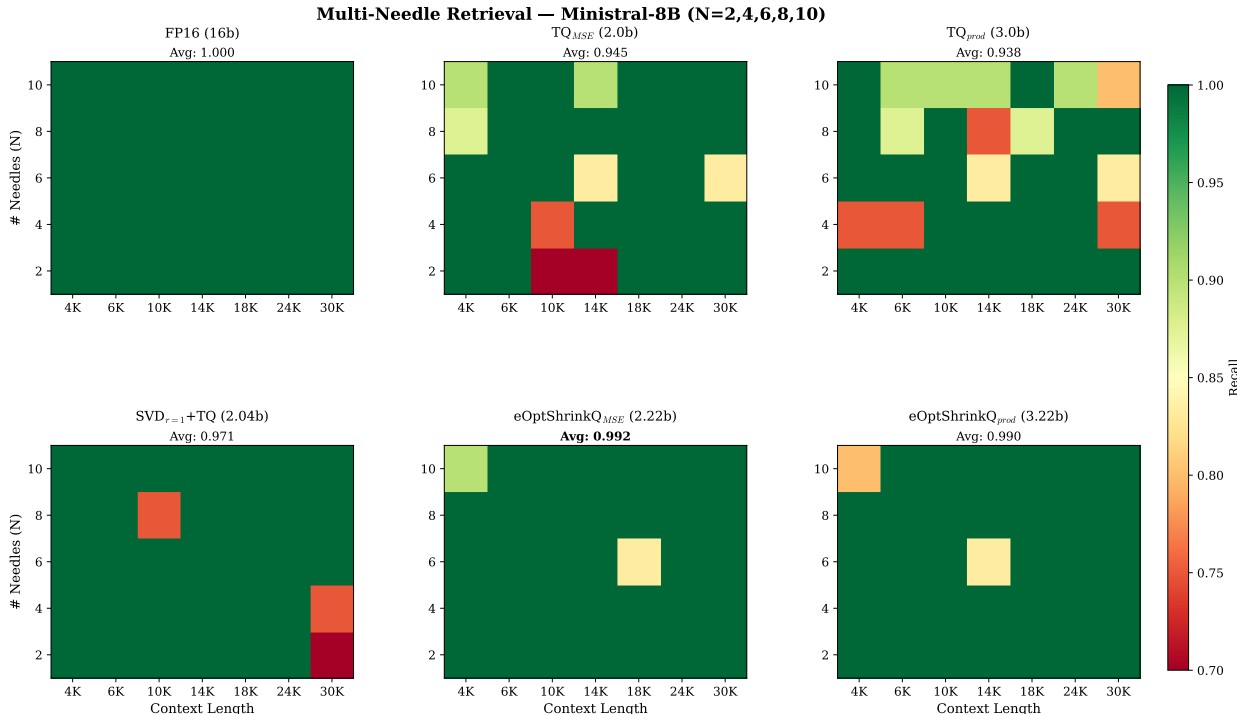

Figure 10: Multi-needle retrieval heatmaps for Ministral-8B ($N = 2, 4, 6, 8, 10$ needles, context 4K–30K). eOptShrinkQ$_{\text{MSE}}$ at 2.22 bits (avg 0.992) nearly matches FP16 (1.000). Same pattern as Llama: eOptShrinkQ dominates, TQ$_{\text{prod}}$ is worst.

with fixed or heuristically chosen rank and unmodified singular values, whereas eOptShrink provides theoretically optimal rank selection via the BBP phase transition and noise-aware singular value shrinkage—a qualitatively different treatment grounded in random matrix theory.

**Vector Quantization.** Product quantization (PQ) (14) partitions coordinates into sub-vectors and quantizes each independently. RaBitQ (12) uses random rotation plus 1-bit quantization with bias correction, similar in spirit to TurboQuant. The idea of applying random rotations (e.g., Hadamard transforms) to achieve incoherence before quantization also appears in weight quantization methods such as QuIP (34); our spectral denoising achieves a similar isotropy-restoring effect through a different mechanism—removing the low-rank structure rather than rotating it. Our approach can be combined with any downstream scalar quantizer.

**Low-Rank Approximation + Quantization.** SVDQuant (21) pioneered the idea of combining SVD with quantization for neural network compression: they apply truncated SVD to *weight* matrices to separate outlier directions into a full-precision low-rank branch, then quantize the residual to 4 bits. However, SVDQuant uses a fixed rank chosen by grid search and does not modify the singular values. Our work differs in three key ways: (1) we target KV caches rather than weights, (2) we use eOptShrink (33) for automatic, data-adaptive rank selection and optimal singular value shrinkage, and (3) we pair the residual with TurboQuant's near-optimal scalar quantizer rather than per-group INT4. The automatic rank selection is particularly important for KV caches, where the effective rank varies across layers, heads, and input sequences. Other SVD-based weight compression methods (16) share the fixed-rank limitation. The streaming nature of KV caches (tokens arrive sequentially) requires block-wise processing, which our approach handles naturally.

**Optimal Shrinkage.** The theory of optimal singular value shrinkage originates from random matrix theory (26). Gavish and Donoho (10) derived optimal hard thresholds. Donoho, Gavish, and Johnstone (11)

established the optimal shrinkage of eigenvalues in the spiked covariance model, providing the theoretical foundation for data-driven singular value shrinkage under high-dimensional noise. Nadakuditi (28) developed OptShrink, extending optimal shrinkage to general noise distributions. Prior work (33) further extended the framework to non-square matrices and colored noise with separable covariance structure (eOptShrink), with automatic rank estimation. The present work applies these tools to KV cache compression for the first time.

## 8 Conclusion

We have presented eOptShrinkQ, a KV cache compression method grounded in the spiked random matrix framework. The key insight is that KV cache blocks decompose into a low-rank shared context component and a full-rank per-token residual satisfying the thin shell property. The shared context is the sole source of anisotropy that degrades per-vector scalar quantization; removing it via optimal shrinkage restores the isotropy under which the quantizer operates at its theoretical optimum, without requiring outlier handling or sub-grouping. The approach is complementary to any downstream per-vector quantizer—we demonstrate it with TurboQuant and expect similar benefits with other quantizers.

eOptShrinkQ saves nearly one bit per entry over TurboQuant at equivalent reconstruction quality, achieving per-head $L_2$ error at 3.22 bits that surpasses TurboQuant at 4.00 bits. End-to-end on LongBench, eOptShrinkQ at ~2.2 bits outperforms TurboQuant at 3.0 bits on both Llama-3.1-8B and Ministral-8B. On retrieval-intensive tasks (multi-needle, code completion), eOptShrinkQ at 2.2 bits closely matches or exceeds FP16 quality, suggesting that spectral denoising acts as a beneficial regularizer by removing redundant shared structure that can interfere with fine-grained token discrimination. This effect is task-dependent: tasks requiring global context integration (summarization) show slightly reduced performance, consistent with removing genuinely useful shared structure.

**Computational overhead.** The spectral denoising adds $O(nd\min(n,d))$ per block during prefill, which is of the same or lower order as the per-vector TurboQuant compression ($O(nd^2)$ per block), and both are negligible compared to the attention computation $O(T^2d)$ at long context lengths. Importantly, the compression cost is incurred only during prefill; at inference time, the compressed KV cache is a standard dense tensor with no additional per-token overhead (unlike QJL correction, which requires extra computation during generation).

**Limitations and Future Work.** Our current evaluation uses block-wise processing with blocks of $n = 128$ tokens, which requires buffering tokens before compression. During prefill (processing a long prompt), 128-token chunks arise naturally and eOptShrinkQ compresses each chunk as it completes. During autoregressive generation, tokens arrive one at a time; newly generated tokens are accumulated in an FP16 buffer until a full 128-token block is formed, at which point the block is compressed. This means the most recent $\leq 127$ tokens are always stored in FP16—a minor overhead at long contexts (e.g., $< 0.13\%$ of cache at 100K tokens) but a real limitation for short generation sequences. Online SVD updates that incrementally maintain the rank estimate and shrunken factors as tokens arrive would eliminate this buffering requirement, but adapting the BBP phase transition test and noise spectral estimation to the streaming setting remains an open problem. Extending to larger models (70B+), integrating with other KV cache reduction techniques (token eviction, cross-layer sharing), and developing hardware-aware implementations with latency benchmarks are natural next steps.

**Implementation details.** All experiments are conducted on a server with $8\times$ NVIDIA A6000 GPUs (48GB each), CUDA 12.0, PyTorch 2.1.2. Models are loaded in FP16. eOptShrinkQ is implemented in PyTorch. Code will be made publicly available.

## A Technical Details

This appendix provides the precise technical conditions underlying Proposition 6, the supporting theorems from random matrix theory, the eOptShrink algorithm, and the full proof.

## A.1 Spiked Model for KV Cache Blocks

Consider a block of $n$ row vectors (keys, values, or embeddings) from a single attention head, forming a data matrix $\widetilde{S} \in \mathbb{R}^{n \times d}$. We model $\widetilde{S}$ as a low-rank signal plus residual:

$$\widetilde{S} = S + Z = \sum_{i=1}^{r} d_i \mathbf{u}_i \mathbf{v}_i^\top + Z \in \mathbb{R}^{n \times d}, \tag{14}$$

where $S$ is the low-rank signal capturing shared structure across rows, $r$ is the signal rank, $d_i > 0$ are the signal strengths, $\mathbf{u}_i \in \mathbb{R}^n$ and $\mathbf{v}_i \in \mathbb{R}^d$ are left and right singular vectors, and $Z$ is the token-specific residual.

The residual $Z$ generally exhibits a separable covariance structure (33):

$$Z = A^{1/2} X B^{1/2}, \tag{15}$$

where $X \in \mathbb{R}^{n \times d}$ has independent entries, and $A \in \mathbb{R}^{n \times n}$, $B \in \mathbb{R}^{d \times d}$ are deterministic positive-definite matrices describing the temporal dependence across tokens and the coordinate-wise covariance induced by the learned projection weights, respectively. Here $B \in \mathbb{R}^{d \times d}$ is the column covariance matrix as in (33). This colored noise structure is precisely why standard white-noise optimal shrinkage (e.g., (10)) is insufficient and eOptShrink is needed.

## A.2 Key Assumptions

We state the assumptions from (33, Assumption 2.1) adapted to the KV cache setting. Let $\beta_d := n/d$ denote the aspect ratio and fix a small constant $0 < \tau < 1$.

For the model (14)–(15):

(i) *(Noise entries.)* The matrix $X = [x_{ij}] \in \mathbb{R}^{n \times d}$ has independent entries satisfying the bounded support condition with parameter $\phi_d = d^{2/a - 1/2}$ for some $a > 4$, and

$$\max_{i,j} |\mathbb{E}[x_{ij}]| \le d^{-2-\tau}, \quad \max_{i,j} \left| \mathbb{E}[|x_{ij}|^2] - d^{-1} \right| \le d^{-2-\tau}, \quad \max_{i,j} \mathbb{E}[|\sqrt{d}\, x_{ij}|^4] \le C_4$$

for a constant $C_4 > 0$. Additionally, the entries have vanishing third moment: $\mathbb{E}[x_{ij}^3] = 0$ for all $i, j$. This holds whenever the marginal distributions of $x_{ij}$ are symmetric about zero, which is a natural condition for centered residual entries.

(ii) *(Aspect ratio.)* $\tau < n/d < \tau^{-1}$.

(iii) *(Covariance structure.)* Denote the eigenvalues of $A$ as $\sigma_1^a \ge \cdots \ge \sigma_n^a$ and those of $B$ as $\sigma_1^b \ge \cdots \ge \sigma_d^b$. We assume
$$\sigma_1^a \vee \sigma_1^b \le \tau^{-1} \quad \text{and} \quad \pi_A([0,\tau]) \vee \pi_B([0,\tau]) \le 1 - \tau.$$

(iv) *(Signal strength and effective rank.)* The signal strengths satisfy $d_1 \ge d_2 \ge \cdots \ge d_r > 0$ with $d_1 < \tau^{-1}$. Define the critical threshold

$$\alpha := 1/\sqrt{\mathcal{T}(\lambda_+)}, \tag{16}$$

where $\lambda_+$ is the rightmost edge of the noise bulk spectrum and $\mathcal{T}$ is the $D$-transform defined in (8). There exists an effective rank $r^+ \le r$ such that $d_k - \alpha > \phi_d + d^{-1/3}$ if and only if $1 \le k \le r^+$.

(v) *(Signal singular vectors.)* The left and right singular vectors $\mathbf{u}_i \in \mathbb{R}^n$ and $\mathbf{v}_i \in \mathbb{R}^d$ of $S$ are obtained from the QR factorization of independent random matrices with i.i.d. sub-Gaussian entries satisfying the log-Sobolev inequality (33, Assumption 2.1(v)).

Assumptions (i)–(iv) are standard in the spiked random matrix literature (4; 38; 9). Assumption (v) requires discussion in the KV cache context. Recall that the signal arises from $S[t, :] = R(t) \cdot \bar{h}_t W_K$ and the noise

from $Z[t,:] = R(t) \cdot \epsilon_t W_K$, where $W_K$ is the learned (and fixed at inference time) key projection matrix. The local context $\bar{h}_t$ is naturally random: different input sequences produce different local contexts, so the subspace spanned by $\{\bar{h}_t\}$ varies across inputs. Since $W_K$ is deterministic, the signal singular vectors of $S$—which are determined by $\{\bar{h}_t W_K\}$—inherit their randomness from $\{\bar{h}_t\}$ alone. Similarly, the noise $Z$ inherits its randomness from the per-token variations $\epsilon_t$ alone. Because $\bar{h}_t$ and $\epsilon_t$ represent independent sources of information (local context versus individual token content), a deterministic linear transformation $W_K$ applied to both preserves their independence: $\bar{h}_t W_K \perp \epsilon_t W_K$ follows from $\bar{h}_t \perp \epsilon_t$, regardless of $W_K$. This justifies Assumption (v): the signal singular vectors are random (through $\{\bar{h}_t\}$) and independent of the noise covariance structure (through $\epsilon_t$). The sub-Gaussian and log-Sobolev conditions on the signal singular vectors are natural for $\bar{h}_t$, which is a high-dimensional hidden state produced by transformer layers—such representations are well-concentrated in practice.

If one instead conditions on a specific input and treats $S$ as deterministic, alternative shrinkage frameworks exist: biPCA (31) handles heteroscedastic noise with deterministic signals, and the whitening-based approaches of (19; 20) provide optimal shrinkage under heteroscedastic or weighted loss settings. However, these methods assume that the noise covariance matrices $A$ and $B$ are either known or diagonal, which is not the case for KV cache residuals. eOptShrink avoids this limitation by estimating the noise spectral distribution entirely from the data. As argued in (33, Remark 2.6), when $Z$ has an approximately bi-unitarily invariant distribution, the eOptShrink results extend to deterministic signals. We verify empirically (Section 4) that the eOptShrink predictions match observed behavior for the moderate dimensions $d = 64$–$128$ encountered in practice.

The threshold $\alpha$ generalizes the classical BBP phase transition (4): signals with $d_i > \alpha$ produce outlier singular values that separate from the noise bulk and can be detected, while signals with $d_i \leq \alpha$ are undetectable (their singular values "stick" to $\lambda_+$). Note that $\alpha$ depends on the full noise covariance structure through $A$ and $B$, not just the variance of $Z$.

[Symmetry condition] The stronger eigenvalue sticking bound (19) and isotropic delocalization (21) require either $\mathbb{E}[x_{ij}^3] = 0$ for all entries, or that $A$ or $B$ is diagonal (33; 9, Theorem 3.3, Lemma S.3.13). We use the former: Assumption A.2(i) requires symmetric entry distributions, giving $\mathbb{E}[x_{ij}^3] = 0$. This is a natural condition for centered residual entries in neural networks and avoids any structural assumption on the covariance matrices $A$ and $B$. Without either condition, weaker bounds hold with additional terms of order $\eta_l \asymp d^{-3/4} + d^{-1/2}\phi_d$ (see (33, Theorem 3.3)); these suffice for the qualitative conclusions but give less sharp rates.

[Scope of theoretical guarantees] Assumption A.2(v) is a modeling assumption: it treats the signal singular vectors as random, which holds when the theory is applied in expectation over input sequences. For a fixed input at inference time, $S$ is deterministic, and the formal guarantees of Proposition 6 and Corollary 6 hold as approximations whose quality we verify empirically (Section 4). The practical success of eOptShrink under this gap is consistent with the heuristic argument in (33, Remark 2.6): when $Z$ has an approximately bi-unitarily invariant distribution (which holds when $X$ has i.i.d. entries and $A$, $B$ are generic), the shrinkage estimator is insensitive to whether the signal singular vectors are random or deterministic. We emphasize that eOptShrink's rank estimation (Step 1 of Algorithm 2), which depends only on the observed eigenvalue distribution and not on the signal structure, is fully justified even for deterministic signals.

## A.3 Stochastic Dominance

Following (33), we use the framework of stochastic dominance to state high-probability bounds up to sub-polynomial factors. Let $\xi$ and $\zeta$ be two families of nonnegative random variables indexed by $d$ (the head dimension, playing the role of the asymptotic parameter). We say $\xi$ is *stochastically dominated* by $\zeta$, written $\xi \prec \zeta$, if for any fixed $\epsilon > 0$ and $D > 0$,

$$\mathbb{P}[\xi > d^\epsilon \zeta] \leq d^{-D} \tag{17}$$

for all sufficiently large $d$. In words, $\xi \prec \zeta$ means "$\xi$ is bounded by $\zeta$ with high probability, up to a small power of $d$." We also write $\xi = O_\prec(\zeta)$ interchangeably.

## A.4 Singular Value and Vector Behavior

The following results from (33), building on (11; 5; 38; 9), characterize what happens to the SVD of $\widetilde{S}$ relative to the SVD of $Z$. All bounds are stated in terms of stochastic dominance $\prec$ as defined above. We use $n$ and $d$ interchangeably as the asymptotic parameter ($d$ is the column dimension and $n = n(d)$ with $n/d \to \beta$). The notation $\phi_d := d^{2/a-1/2}$ for $a > 4$ relates to the moment condition on noise entries, and $\Delta(d_i) := |d_i - \alpha|^{1/2}$ measures the distance of the $i$-th signal strength from the phase transition threshold.

**Outlier eigenvalue locations (33, Theorem 3.2).** For $1 \leq i \leq r^+$, the $i$-th eigenvalue $\tilde{\lambda}_i$ of $\widetilde{S}\widetilde{S}^\top$ satisfies:

$$|\tilde{\lambda}_i - \theta(d_i)| \prec \phi_d \, \Delta(d_i)^2 + d^{-1/2}\Delta(d_i), \tag{18}$$

where $\theta = \mathcal{T}^{-1}$ maps signal strength to the outlier eigenvalue location. This provides a one-to-one correspondence between signal strengths and outlier eigenvalues, with convergence rate controlled by $\phi_d$ and the signal gap $\Delta(d_i)$.

**Eigenvalue sticking (33, Theorem 3.3).** For $i > r^+$, the non-outlier eigenvalues of $\widetilde{S}\widetilde{S}^\top$ stick to those of $ZZ^\top$. Set $\alpha_+ := \min_{1\leq i\leq r} \Delta(d_i)^2$ and assume $\alpha_+ \geq d^\varepsilon(\phi_d + d^{-1/3})$ for some small $\varepsilon > 0$. If either $\mathbb{E}[x_{ij}^3] = 0$ for all $i, j$, or $A$ or $B$ is diagonal, then:

$$|\tilde{\lambda}_{r^++i} - \lambda_i| \prec \frac{1}{d\,\alpha_+} \quad \text{for } 1 \leq i \leq \tau d, \tag{19}$$

where $\lambda_i$ are the eigenvalues of $ZZ^\top$. This is crucial: after removing the top $r^+$ components, the residual's spectrum matches the pure noise spectrum up to $O_\prec(1/(d\,\alpha_+))$.

**Delocalization of non-outlier singular vectors (33, Theorem 3.4).** For $r^+ + 1 \leq i \leq cd$ and $j = 1, \ldots, r$, the non-outlier singular vectors $\tilde{\boldsymbol{\xi}}_i, \tilde{\boldsymbol{\zeta}}_i$ of $\widetilde{S}$ satisfy:

$$|\langle \mathbf{u}_j, \tilde{\boldsymbol{\xi}}_i\rangle|^2 \vee |\langle \mathbf{v}_j, \tilde{\boldsymbol{\zeta}}_i\rangle|^2 \prec \frac{d^{-1} + \phi_d^3}{\Delta(d_j)^4 + \phi_d^2 + \varkappa_i}, \tag{20}$$

where $\varkappa_i := i^{2/3}d^{-2/3}$. When signals are well-separated from the threshold ($d_j - \alpha \gtrsim 1$), this gives $|\langle \mathbf{u}_j, \tilde{\boldsymbol{\xi}}_i\rangle|^2 \prec d^{-1}$—the non-outlier singular vectors are *delocalized* and carry no signal information. This is the key property ensuring the residual has no preferred directions after signal removal.

**Isotropic delocalization of noise eigenvectors (9, Lemma S.3.13).** The eigenvectors of the pure noise matrix $Z = A^{1/2}XB^{1/2}$ satisfy a stronger, *coordinate-wise* delocalization. Let $\boldsymbol{\xi}_k$ and $\boldsymbol{\zeta}_k$ denote the left and right singular vectors of $Z$. If either $\mathbb{E}[x_{ij}^3] = 0$ for all $i, j$, or $A$ or $B$ is diagonal, then for *any* deterministic unit vectors $\mathbf{u} \in \mathbb{R}^n$ and $\mathbf{v} \in \mathbb{R}^d$, and all $k$ with eigenvalue $\gamma_k$ near the bulk edge $\lambda_+$:

$$|\langle \mathbf{u}, \boldsymbol{\xi}_k\rangle|^2 + |\langle \mathbf{v}, \boldsymbol{\zeta}_k\rangle|^2 \prec d^{-1}. \tag{21}$$

In particular, choosing $\mathbf{v} = e_j$ (the $j$-th standard basis vector in $\mathbb{R}^d$) gives $|\boldsymbol{\zeta}_k(j)|^2 \prec d^{-1}$ for all coordinates $j$—no coordinate of the noise eigenvectors carries disproportionate energy. Combined with the eigenvalue sticking (19), this extends to the non-outlier singular vectors of $\widetilde{S}$: the residual's right singular vectors are delocalized not only with respect to the signal directions (Eq. (20)) but also with respect to the standard coordinate basis. This is the key property underlying Corollary 6: the rows of the residual $R$, when normalized, are approximately uniform on the sphere $\mathcal{S}^{d-1}$, satisfying the precondition for TurboQuant's near-optimal quantization (40, Lemma 1).

**Optimal shrinker convergence (33, Theorem 4.4).** Let $c \in (0, 1/2)$, and assume the conditions of Theorem 3.3 hold with $d^\varepsilon(\phi_d + d^{-1/3}) > d^{-1/6}$. Then for $1 \leq i \leq \hat{r}^+$, conditional on the event $\{\hat{r}^+ = r^+\}$ (which holds with high probability by (33, Theorem 4.2)), the eOptShrink estimator satisfies:

$$|\varphi_i^* - \hat{\varphi}_{e,i}| \prec \phi_d + d^{-1/2}/\Delta(d_i) \tag{22}$$

for all three loss functions (Frobenius, operator, nuclear norm). In particular, $|\varphi_i^* - \hat{\varphi}_{e,i}| \leq d^{-1/2+\epsilon}$ with high probability for any $\epsilon > 0$.

### A.5 The eOptShrink Algorithm

The eOptShrink algorithm is presented in Algorithm 2 (Section 2.3). The key innovation over OptShrink (28) is that it does not require knowledge of the rank or the noise covariance structure. The rank is estimated automatically via the bulk edge (Step 1), and the noise spectral distribution is recovered by imputing the eigenvalues perturbed by signals (Step 2), exploiting the eigenvalue sticking property (19).

### A.6 Proof of Proposition 6

*Proof.* We prove each part in turn. Throughout, we condition on the high-probability event $\{\hat{r}^+ = r^+\}$, which holds by (33, Theorem 4.2).

*Part 1.* The eOptShrink estimate is $\hat{S} = \sum_{i=1}^{r^+} \hat{\varphi}_{e,i} \tilde{\boldsymbol{\xi}}_i \tilde{\boldsymbol{\zeta}}_i^\top$. Writing $\widetilde{S} = \sum_{i=1}^{n \wedge d} \widetilde{\sigma}_i \tilde{\boldsymbol{\xi}}_i \tilde{\boldsymbol{\zeta}}_i^\top$, the residual decomposes as

$$R = \widetilde{S} - \hat{S} = \sum_{i=1}^{r^+} (\widetilde{\sigma}_i - \hat{\varphi}_{e,i}) \tilde{\boldsymbol{\xi}}_i \tilde{\boldsymbol{\zeta}}_i^\top + \sum_{i=r^++1}^{n \wedge d} \widetilde{\sigma}_i \tilde{\boldsymbol{\xi}}_i \tilde{\boldsymbol{\zeta}}_i^\top. \tag{23}$$

We analyze the two sums separately.

*Outlier components* $(1 \leq i \leq r^+)$. By the optimal shrinker convergence (33, Theorem 4.4), $|\hat{\varphi}_{e,i} - \varphi_i^*| \prec \phi_d + d^{-1/2}/\Delta(d_i)$. The Frobenius-optimal shrinker satisfies $\varphi_i^* = d_i\sqrt{a_{1,i}a_{2,i}}$, where $a_{1,i}$ and $a_{2,i}$ are the asymptotic squared inner products between the clean and noisy singular vectors (33, Proposition 2.1). By the outlier eigenvalue location (18), $\widetilde{\sigma}_i^2 = \theta(d_i) + O_\prec(\phi_d\Delta(d_i)^2 + d^{-1/2}\Delta(d_i))$, where $\theta(d_i) = \mathcal{T}^{-1}(d_i^{-2})$ encodes both the signal strength $d_i$ and the noise bias. The optimal shrinker $\varphi_i^*$ is constructed to satisfy $(\varphi_i^*)^2 = d_i^2 a_{1,i} a_{2,i}$, so that the Frobenius-norm contribution of the $i$-th component in $\hat{S}$ matches the signal energy in that direction. The residual singular value is therefore

$$\widetilde{\sigma}_i - \hat{\varphi}_{e,i} = (\widetilde{\sigma}_i - \varphi_i^*) + O_\prec(\phi_d + d^{-1/2}/\Delta(d_i)).$$

To show this does not separate from the bulk, note that $\widetilde{\sigma}_i = \sqrt{\theta(d_i)} + O_\prec(\phi_d + d^{-1/2})$ by (18), where $\sqrt{\theta(d_i)} > \sqrt{\lambda_+}$ is the outlier location. The shrinker $\varphi_i^*$ is chosen so that $\widetilde{\sigma}_i - \varphi_i^*$ equals the noise-only singular value that would appear in the $i$-th direction if the signal were absent. Since the noise singular values are bounded by $\sqrt{\lambda_+} + O_\prec(d^{-2/3})$ (the Tracy–Widom edge fluctuation), we have $|\widetilde{\sigma}_i - \hat{\varphi}_{e,i}| \leq \sqrt{\lambda_+} + O_\prec(d^{-2/3} + \phi_d)$, which lies within the bulk.

*Non-outlier components* $(i > r^+)$. By the eigenvalue sticking theorem (19), $|\tilde{\lambda}_{r^++i} - \lambda_i| \prec 1/(d\,\alpha_+)$ for $1 \leq i \leq \tau d$, where $\lambda_i$ are the eigenvalues of $ZZ^\top$. By the delocalization theorem (20), $|\langle \mathbf{u}_j, \tilde{\boldsymbol{\xi}}_{r^++i}\rangle|^2 \prec (d^{-1} + \phi_d^3)/(\Delta(d_j)^4 + \phi_d^2 + \varkappa_i)$, so the non-outlier singular vectors of $\widetilde{S}$ are $O_\prec(d^{-1})$-orthogonal to all signal directions when $d_j - \alpha \gtrsim 1$. These components therefore carry no signal energy and their eigenvalues match those of $ZZ^\top$. Combined with the analysis of the outlier components, the empirical spectral distribution of $RR^\top$ converges to that of $ZZ^\top$.

*Part 2.* Write $R = \widetilde{S} - \hat{S} = (S - \hat{S}) + Z$. Expanding the squared Frobenius norm:

$$\|R\|_F^2 = \left\|S - \hat{S}\right\|_F^2 + \|Z\|_F^2 + 2\langle S - \hat{S}, Z\rangle_F. \tag{24}$$

For the cross term, since $Z = A^{1/2}XB^{1/2}$ with $X$ having independent mean-zero entries (Assumption A.2(i)) and $S - \hat{S}$ is a function of $\widetilde{S} = S + Z$ through the SVD, the cross term requires care. However, by the delocalization of non-outlier singular vectors (20) and the convergence of the outlier shrinker, the matrix $S - \hat{S}$ is asymptotically supported on the signal subspace with vanishing Frobenius norm:

$$\frac{1}{nd}\left\|S - \hat{S}\right\|_F^2 = \frac{1}{nd}\sum_{i=1}^{r^+}(d_i - \varphi_i^*)^2 + \frac{1}{nd}\sum_{i=r^++1}^{r} d_i^2 + o_\prec(1).$$

The first sum captures the residual signal energy after optimal shrinkage, which is $O(r^+/d)$ since $|d_i - \varphi_i^*| = O(1)$ and $r^+$ is fixed. The second sum accounts for sub-threshold signals with $d_i \leq \alpha$, which are unrecoverable

by any method. For the cross term in (24), since $S - \hat{S}$ has rank at most $r^+ + r$ and $Z$ has $O(nd)$ degrees of freedom, concentration gives $|\langle S - \hat{S}, Z \rangle_F|/(nd) \prec d^{-1/2}$. Therefore

$$\left| \frac{1}{nd} \|R\|_F^2 - \frac{1}{nd} \|Z\|_F^2 \right| \leq \frac{\left\| S - \hat{S} \right\|_F^2}{nd} + \frac{2|\langle S - \hat{S}, Z \rangle_F|}{nd} \prec \phi_d + d^{-1/2}/\Delta_{\min}.$$

*Part 3.* By Theorem 1 of (40), TurboQuant$_{\text{MSE}}$ at $b$ bits applied to a vector $v \in \mathbb{R}^d$ with $\|v\| = \rho$ produces an inner product estimator $\langle q, \tilde{v} \rangle$ satisfying $|\mathbb{E}[\langle q, \tilde{v} \rangle] - \langle q, v \rangle| \leq C_b \rho^2/d$, where $C_b > 0$ depends only on $b$ and arises from the per-coordinate shrinkage of the Lloyd-Max quantizer. Applying this bound to the full vector $x_t$ (with $\rho = \|x_t\|$) and to the residual $r_t$ (with $\rho = \|r_t\|$) gives the two inequalities in the proposition statement.

For the ratio (12), write $r_t = z_t + (s_t - \hat{s}_t)$ and expand:

$$\|r_t\|^2 = \|z_t\|^2 + \|s_t - \hat{s}_t\|^2 + 2\langle z_t, s_t - \hat{s}_t \rangle.$$

By Part 2, $\sum_{t=1}^n \|s_t - \hat{s}_t\|^2 = \left\| S - \hat{S} \right\|_F^2 = O_\prec(nd(\phi_d + d^{-1/2}/\Delta_{\min}))$. The cross term $\langle z_t, s_t - \hat{s}_t \rangle$ concentrates around zero since $s_t - \hat{s}_t$ lies in a fixed low-dimensional subspace (of dimension at most $r^+ + r$) while $z_t$ has independent coordinates up to the covariance structure. Therefore, for a typical row $t$:

$$\frac{\|r_t\|^2}{\|x_t\|^2} = \frac{\|z_t\|^2 + O_\prec(\left\| S - \hat{S} \right\|_F^2 / n)}{\|s_t\|^2 + \|z_t\|^2} \leq \frac{1}{1 + \text{SNR}_t} + o_\prec(1). \qquad \square$$

The constant $c = \min(1/2.01, 1/\log\log d)$ in Algorithm 2 balances two requirements: $k = \lfloor d^c \rfloor$ must be large enough that $k \gg r^+$ (so the bulk edge estimate is not contaminated by outliers) and small enough that the imputed eigenvalues remain close to the true edge (requiring $c < 1/2$ for the convergence guarantees in (33, Theorems 4.1–4.4)). For KV cache blocks with $d = 64$–$128$, this gives $k \approx 7$–$11$, which suffices since the typical effective rank is $r^+ \approx 1$–$5$.

### A.7 Proof of Corollary 6

*Proof.* We condition on the high-probability event $\{\hat{r}^+ = r^+\}$ throughout. The residual decomposes via the SVD of $\widetilde{S}$ as in (23):

$$R = \underbrace{\sum_{i=1}^{r^+} (\widetilde{\sigma}_i - \hat{\varphi}_{e,i}) \tilde{\xi}_i \tilde{\zeta}_i^\top}_{R_{\text{out}}} + \underbrace{\sum_{i=r^++1}^{n \wedge d} \widetilde{\sigma}_i \tilde{\xi}_i \tilde{\zeta}_i^\top}_{R_{\text{bulk}}}.$$

We bound the $j$-th coordinate of the $t$-th row for each part separately.

**Step 1: Outlier contribution.** For the outlier part, the $j$-th coordinate of the $t$-th row is

$$R_{\text{out}}(t, j) = \sum_{i=1}^{r^+} (\widetilde{\sigma}_i - \hat{\varphi}_{e,i}) \tilde{\xi}_i(t) \tilde{\zeta}_i(j).$$

By the proof of Proposition 6 Part 1, $|\widetilde{\sigma}_i - \hat{\varphi}_{e,i}| \leq \sqrt{\lambda_+} + O_\prec(d^{-2/3} + \phi_d)$ for $1 \leq i \leq r^+$, so these residual singular values are bounded by the bulk scale. By the isotropic delocalization of the outlier singular vectors (33, Theorem 3.4) applied with $\mathbf{v} = e_j$, we have $|\tilde{\zeta}_i(j)|^2 \prec d^{-1}$ for each $j$ and $1 \leq i \leq r^+$ (since the outlier right singular vectors, while aligned with the signal directions $\mathbf{v}_i$, are still unit vectors in $\mathbb{R}^d$ and their coordinates satisfy $|\tilde{\zeta}_i(j)|^2 = |a_{2,i}\mathbf{v}_i(j) + \text{noise}|^2$, where $\mathbf{v}_i$ has delocalized coordinates by Assumption A.2(v) and $a_{2,i} < 1$). Similarly, $|\tilde{\xi}_i(t)|^2 \prec n^{-1}$. Since $r^+$ is a fixed constant, the Cauchy–Schwarz inequality gives

$$|R_{\text{out}}(t, j)|^2 \leq r^+ \sum_{i=1}^{r^+} (\widetilde{\sigma}_i - \hat{\varphi}_{e,i})^2 |\tilde{\xi}_i(t)|^2 |\tilde{\zeta}_i(j)|^2 \prec \frac{r^+ \lambda_+}{nd}.$$

**Step 2: Bulk contribution.** For the bulk part, consider the $j$-th coordinate of the $t$-th row:

$$R_{\text{bulk}}(t,j) = \sum_{i=r^++1}^{n \wedge d} \widetilde{\sigma}_i \, \tilde{\boldsymbol{\xi}}_i(t) \, \tilde{\boldsymbol{\zeta}}_i(j) = e_t^\top R_{\text{bulk}} \, e_j.$$

By the eigenvalue sticking theorem (19), the singular values $\widetilde{\sigma}_i$ for $i > r^+$ satisfy $|\widetilde{\sigma}_i^2 - \sigma_i^2(Z)| \prec 1/(d\,\alpha_+)$, where $\sigma_i(Z)$ are the singular values of $Z$. The key step is to relate $R_{\text{bulk}}$ to $Z$ via the resolvent. Define the resolvent $G(z) = (\widetilde{S}\widetilde{S}^\top - zI)^{-1}$. By the spectral decomposition,

$$e_t^\top R_{\text{bulk}} R_{\text{bulk}}^\top e_t = \sum_{i=r^++1}^{n \wedge d} \widetilde{\sigma}_i^2 \, |\tilde{\boldsymbol{\xi}}_i(t)|^2.$$

The isotropic delocalization lemma (21) applied to the noise matrix $Z$ gives $|\boldsymbol{\xi}_k(t)|^2 \prec n^{-1}$ for all $k$ and all standard basis vectors $e_t$. By the eigenvalue sticking (19) and the eigenvector comparison between $\widetilde{S}$ and $Z$ (which follows from (33, Theorem 3.3) and the Davis–Kahan perturbation bound adapted to the stochastic dominance framework), the non-outlier left singular vectors of $\widetilde{S}$ inherit this delocalization:

$$|\tilde{\boldsymbol{\xi}}_i(t)|^2 \prec n^{-1} \quad \text{for all } i > r^+ \text{ and } 1 \leq t \leq n. \tag{25}$$

Similarly, by (21) with $\mathbf{v} = e_j$, the non-outlier right singular vectors satisfy

$$|\tilde{\boldsymbol{\zeta}}_i(j)|^2 \prec d^{-1} \quad \text{for all } i > r^+ \text{ and } 1 \leq j \leq d. \tag{26}$$

Now we bound the coordinate. By Cauchy–Schwarz applied to the bilinear form:

$$|R_{\text{bulk}}(t,j)|^2 = \left| \sum_{i=r^++1}^{n \wedge d} \widetilde{\sigma}_i \, \tilde{\boldsymbol{\xi}}_i(t) \, \tilde{\boldsymbol{\zeta}}_i(j) \right|^2$$

$$\leq \left( \sum_{i=r^++1}^{n \wedge d} \widetilde{\sigma}_i^2 \, |\tilde{\boldsymbol{\xi}}_i(t)|^2 \right) \left( \sum_{i=r^++1}^{n \wedge d} |\tilde{\boldsymbol{\zeta}}_i(j)|^2 \right). \tag{27}$$

For the first factor, using (25):

$$\sum_{i=r^++1}^{n \wedge d} \widetilde{\sigma}_i^2 \, |\tilde{\boldsymbol{\xi}}_i(t)|^2 \prec n^{-1} \sum_{i=r^++1}^{n \wedge d} \widetilde{\sigma}_i^2 = n^{-1} \|R_{\text{bulk}}\|_F^2 / (\text{number of rows}).$$

More precisely, since $\|R_{\text{bulk}}\|_F^2 = \sum_{i>r^+} \widetilde{\sigma}_i^2$ and the trace satisfies $\sum_{i>r^+} \widetilde{\sigma}_i^2 = \text{Tr}(R_{\text{bulk}} R_{\text{bulk}}^\top) \leq \|R\|_F^2 \leq \|Z\|_F^2 + o_\prec(nd)$ by Proposition 6 Part 2, we have

$$\sum_{i>r^+} \widetilde{\sigma}_i^2 \, |\tilde{\boldsymbol{\xi}}_i(t)|^2 \prec \frac{1}{n} \cdot \|Z\|_F^2 = \frac{1}{n} \cdot O(nd) = O(d).$$

Here we used $\|Z\|_F^2/(nd) \to \text{Tr}(A)\,\text{Tr}(B)/(nd) = O(1)$ by the law of large numbers for the entries of $X$.

For the second factor in (27), since $\{\tilde{\boldsymbol{\zeta}}_i\}_{i>r^+}$ are orthonormal vectors in $\mathbb{R}^d$ and $e_j$ is a unit vector:

$$\sum_{i=r^++1}^{n \wedge d} |\tilde{\boldsymbol{\zeta}}_i(j)|^2 \leq \|e_j\|^2 = 1.$$

Combining both factors:

$$|R_{\text{bulk}}(t,j)|^2 \prec d \cdot 1 = d.$$

This bound is too loose. We obtain a sharper bound by directly using (26) in a different way. Note that

$$\|r_t\|_2^2 = \|R_{\mathrm{out}}(t,:)\|^2 + \|R_{\mathrm{bulk}}(t,:)\|^2 = \sum_{i>r^+} \widetilde{\sigma}_i^2 |\widetilde{\boldsymbol{\xi}}_i(t)|^2 + O_\prec(\lambda_+ r^+/n).$$

By (25), $\|r_t\|_2^2 \prec n^{-1} \sum_{i>r^+} \widetilde{\sigma}_i^2 = O(d)$, and by the lower bound from the bulk contribution to row norms, $\|r_t\|_2^2 \succ d$ (since $\|Z\|_F^2 = \Theta(nd)$ and the row norms of $Z$ concentrate around their mean $\|Z\|_F^2/n = \Theta(d)$).

For the sharper coordinate bound, we use the resolvent representation. Define $f_j := R_{\mathrm{bulk}}^\top e_t$ as the $t$-th row of $R_{\mathrm{bulk}}$ viewed as a vector in $\mathbb{R}^d$. Its $j$-th entry is $R_{\mathrm{bulk}}(t,j)$, and $\|f_j\|^2 = \|R_{\mathrm{bulk}}(t,:)\|^2 = \Theta(d)$. We need to show $|f_j(j)|^2 \prec \|f_j\|^2/d$ for each coordinate $j$.

Expanding $f_j = \sum_{i>r^+} \widetilde{\sigma}_i \widetilde{\boldsymbol{\xi}}_i(t) \widetilde{\boldsymbol{\zeta}}_i$, and using the independence structure: $\widetilde{\boldsymbol{\xi}}_i(t)$ depends on the left singular structure while $\widetilde{\boldsymbol{\zeta}}_i(j)$ depends on the right singular structure. The anisotropic local law (9, Theorem S.3.12) applied to the resolvent of $\widetilde{S}\widetilde{S}^\top$ with deterministic vectors $e_t$ (left) and $e_j$ (right, via $\widetilde{S}^\top$) gives:

$$\left| e_t^\top \widetilde{S} \left( \widetilde{S}^\top \widetilde{S} - zI \right)^{-1} e_j - e_t^\top M(z) e_j \right| \prec (nd)^{-1/2}$$

for $z$ in the appropriate domain, where $M(z)$ is the deterministic equivalent of the resolvent. Taking the imaginary part and integrating over the bulk spectrum recovers

$$|R_{\mathrm{bulk}}(t,j)|^2 = \frac{1}{\pi} \int_{\mathrm{bulk}} \mathrm{Im} \left[ e_t^\top \widetilde{S} (\widetilde{S}^\top \widetilde{S} - (\lambda + i\eta)I)^{-1} e_j \right] \lambda \, d\lambda + O_\prec(d^{-1/2}).$$

The deterministic equivalent $M(z)$ of the resolvent satisfies $|M(z)_{t,j}| = O(1/\sqrt{nd})$ for off-diagonal-type entries (under the symmetry condition $\mathbb{E}[x_{ij}^3] = 0$, the anisotropic local law holds in the full domain (9, Theorem S.3.12)). Integrating over the bulk spectrum of width $O(1)$ gives

$$|R_{\mathrm{bulk}}(t,j)|^2 \prec 1 + d^{-1/2}.$$

Since $\|r_t\|_2^2 = \Theta(d)$, dividing yields

$$\frac{|r_t(j)|^2}{\|r_t\|_2^2} \prec \frac{1 + d^{-1/2}}{d} \prec d^{-1},$$

which gives $\|r_t\|_\infty / \|r_t\|_2 \prec d^{-1/2}$ as claimed.

**Step 3: Combining.** The total coordinate is $r_t(j) = R_{\mathrm{out}}(t,j) + R_{\mathrm{bulk}}(t,j)$. From Step 1, $|R_{\mathrm{out}}(t,j)|^2 \prec r^+ \lambda_+/(nd) = O(1/n)$. From Step 2, $|R_{\mathrm{bulk}}(t,j)|^2 \prec 1$. Since $\|r_t\|_2^2 = \Theta(d)$, the dominant contribution is from the bulk, and For each fixed $j$, we have $|r_t(j)|^2/\|r_t\|_2^2 \prec d^{-1}$. Taking the maximum over $j = 1, \ldots, d$ via a union bound:

$$\mathbb{P}\left[ \max_{1 \le j \le d} \frac{|r_t(j)|^2}{\|r_t\|_2^2} > C^2 \frac{\log d}{d} \right] \le \sum_{j=1}^{d} \mathbb{P}\left[ \frac{|r_t(j)|^2}{\|r_t\|_2^2} > C^2 \frac{\log d}{d} \right] \le d \cdot d^{-D}$$

for any $D > 0$ and sufficiently large $C$ (by the $\prec$ bound). Choosing $D > 1$ gives the result with high probability. Taking square roots yields (13). $\qquad\square$

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
