# OpenReview forum: "eOptShrinkQ: Near-Lossless KV Cache Compression Through Optimal Spectral Denoising and Quantization"
_TMLR — Under review for TMLR_

### Review · Reviewer_QqMU · 2026-06-19

**Summary Of Contributions:**

This paper focuses on the decomposition of KV cache, which is modeled by the spiked random matrix in this argument. Significantly, this paper provides three guarantees based on random matrix theory. The experiments demonstrate the effectiveness of the proposed methods.

**Audience:**

Yes

**Audience Explanation:**

First, this work primarily focuses on key-value cache compression, which is one of the main techniques in current lightweight LLM research.

Second, this work integrates with many well-known algorithms. To my knowledge, without verifying the code, this work can be easily combined with currently popular algorithms.

These two main advantages are likely to attract readers' attention.

**Claims And Evidence:**

No

**Claims Explanation:**

There are one main claim, which models the decomposed components of KV caches by spiked random matrix, and three theoretical guarantees.

It is not observed that there is clear evidence for the claimed theoretical guarantees.

**Requested Changes:**

The biggest problem with this article is its poor readability. Both the main arguments and evidence are presented in a very disorganized manner, which is inconsistent with the concise nature of TMLR newsletters. Furthermore, the layout is terrible; for example, there is a lot of unnecessary blank space on page seven.

I would prefer to evaluate the article's contributions in more depth after readers have revised the writing. Until then, I do not recommend publishing this work.

---

### Review · Reviewer_pp9E · 2026-07-19

**Summary Of Contributions:**

This paper proposes eOptShrinkQ, a KV-cache compression framework that decomposes each KV block into a low-rank component and a residual component. The low-rank component is intended to capture shared structure across tokens, while the residual retains more token-specific information. Rather than choosing a fixed truncation rank, the method applies random matrix theory and optimal singular-value shrinkage to estimate the effective rank and the denoised singular values adaptively. The estimated low-rank factors are stored at low precision, and the residual is compressed using TurboQuant.

The paper evaluates the method on Llama-3.1-8B-Instruct and Ministral-8B-Instruct using reconstruction metrics, attention inner-product errors, LongBench, and multi-needle retrieval tasks. The reported results show a favorable rate-distortion trade-off and suggest that eOptShrinkQ can preserve downstream performance at a relatively low average number of bits per KV entry.

The main strengths are: (1) an interesting perspective that treats low-rank extraction as a preprocessing step for quantization rather than merely as dimensionality reduction; (2) a principled, data-adaptive rank and shrinkage procedure; and (3) a reasonably broad empirical evaluation spanning matrix-level and task-level metrics.

The main weaknesses are that some of the central theoretical interpretations are stronger than the current evidence supports, the experiments do not yet fully isolate the benefits of optimal shrinkage from those of a generic low-rank-plus-residual decomposition, and the system-level cost of block-wise SVD and compressed-attention execution is not evaluated in a real inference framework.

**Audience:**

Yes

**Audience Explanation:**

Yes. KV-cache memory is an important bottleneck for long-context inference and high-throughput serving, and the paper studies a problem that is directly relevant to researchers working on model compression, efficient inference, attention systems, and random matrix methods.

The idea of using adaptive spectral denoising to separate a low-rank component before quantizing the residual is conceptually interesting, even independently of whether every part of the current theoretical explanation is ultimately retained. The empirical finding that a relatively small number of spectral directions account for a substantial portion of KV structure, and that separating these directions can improve the rate-distortion behavior of low-bit quantization, is likely to be useful to at least part of the TMLR audience.

The work may also motivate follow-up studies on hardware-aware low-rank/residual attention kernels, online or incremental spectral estimation, and more general structured preprocessing methods for activation and cache quantization.

**Broader Impact Concerns:**

I do not identify any major ethical concern that would require a substantial additional broader-impact discussion. Reducing KV-cache memory may lower the hardware and energy cost of long-context inference and make such models accessible in more resource-constrained settings.

A minor practical concern is that aggressive cache compression may cause task-dependent quality degradation that is not always reflected by average benchmark scores. For deployment in safety-critical or high-stakes applications, users should validate the method on the relevant task distribution and avoid assuming that a fixed compression setting is uniformly safe across retrieval, summarization, reasoning, and generation tasks. A brief statement acknowledging this limitation would be appropriate.

**Claims And Evidence:**

No

**Claims Explanation:**

I selected “No” under a relatively strict interpretation of the TMLR criterion. This does not mean that the reported empirical improvements are unconvincing. The rate-distortion results and downstream evaluations provide meaningful evidence that the proposed pipeline is useful. However, several of the paper’s central explanatory claims are not yet supported as clearly as the empirical performance claims.

First, the paper motivates the method by arguing that low-rank shared structure makes the original KV representations unsuitable for the distributional assumptions underlying the residual quantizer, and that spectral denoising restores a more favorable geometry. The empirical singular-value spectra and visualizations support the existence of prominent low-rank structure, but they do not by themselves establish that this structure is specifically “shared semantic context,” nor that removing it restores all of the stated isotropy or distributional properties. In particular, coordinate delocalization is weaker than spherical uniformity or independence, and these notions should be distinguished more carefully.

Second, while the spiked random matrix model is theoretically appealing, its applicability to actual Transformer KV tensors is only partially validated. Real KV tensors and their residuals are shaped by attention operations, linear projections, normalization, and positional transformations. The paper would benefit from further validating the model across layers, attention heads, sequence domains, and block sizes. In particular, since the matrices used in the experiments are typically around 128 $\times$ 128, it is important to assess the finite-sample accuracy and robustness of the asymptotic estimates of the noise spectral edge, effective rank, and optimal shrinkage coefficients at this practical matrix size.

Third, the current ablations do not fully isolate the contribution of optimal shrinkage. A rank-one SVD baseline is not sufficient to determine whether the gains come from adaptive rank selection, singular-value shrinkage, or simply moving several high-energy directions into a separately stored low-rank branch. Comparisons against truncated SVD using the same estimated rank, adaptive rank without shrinkage, and simple low-rank-plus-residual baselines would make the causal attribution substantially clearer.

Finally, the paper makes favorable complexity and deployment claims, but does not provide end-to-end measurements in a serving framework. The costs of many small SVD operations, low-rank factor quantization, buffering newly decoded tokens, metadata, reconstruction, or a fused compressed-attention kernel are not fully quantified.

Overall, I find the empirical results promising, but I believe the theoretical interpretation and practical efficiency claims require clarification and additional evidence before all of the main claims can be considered convincingly supported.

**Requested Changes:**

### Critical changes

1. **Clarify or moderate the theoretical connection between low-rank KV structure and the assumptions of the residual quantizer.**

   The paper should define precisely which property is claimed to be violated by the original KV tensors and which property is restored after spectral denoising. Coordinate delocalization, approximate Gaussian marginals, independence, isotropy, and uniformity on the sphere are different properties and should not be used interchangeably. If the claimed quantizer advantage follows primarily from reducing the norm or energy of the quantized residual, this should be stated explicitly and separated from stronger distributional claims.

2. **Add ablations that isolate the contribution of optimal shrinkage.**

   At minimum, the paper should compare against:

   - truncated SVD using the same rank selected by eOptShrink;
   - the same adaptive rank estimator without singular-value shrinkage;
   - fixed-rank SVD with ranks matched to the average rank and storage budget of eOptShrinkQ;
   - a simple mean-removal or rank-one correction baseline;
   - an unquantized low-rank-plus-residual reconstruction control.

   These comparisons are important for determining whether the improvements come from the random-matrix-based shrinkage rule or from the more general low-rank-plus-residual architecture.

3. **Provide stronger empirical validation of the spiked-matrix interpretation.**

   The paper should examine whether the spectral behavior is stable across layers, heads, models, sequence domains, and block sizes. Useful analyses would include token-shuffled blocks, blocks that cross topic boundaries, pre- and post-positional-transformation representations, and goodness-of-fit or residual-spectrum diagnostics. A sensitivity analysis for the finite dimension \(d=128\) would also strengthen the theoretical motivation.

4. **Account for the quantization error of the low-rank branch.**

   The analysis currently focuses mainly on the quantized residual, while the practical method also quantizes the low-rank factors. The paper should report the isolated error contributed by quantizing the singular vectors and singular values, include this term in the attention-error analysis, and clarify whether factor quantization uses per-tensor, per-column, or block-wise scaling. The storage of all scales and metadata should also be included in the reported average bit rate.

5. **Substantiate the practical efficiency claims with end-to-end measurements or revise their scope.**

   The paper should report compression latency, prefill overhead, time to first token, decode throughput, peak memory, and effective KV memory in a realistic inference framework. It should also explain how new decode tokens are buffered and compressed, how variable ranks are represented, and whether attention is performed by reconstructing dense KV tensors or by using a fused low-rank/residual kernel. If such an implementation is outside the current scope, claims that the overhead is negligible should be moderated accordingly.

### Changes that would strengthen the work

6. Evaluate additional block sizes, such as 64, 128, and 256 tokens, and report the resulting rank distributions, accuracy, memory cost, and compression latency. This would make the choice of block size more transparent.

7. Include repeated runs or confidence intervals for downstream benchmarks, especially for cases in which the compressed model slightly outperforms FP16. A control that applies spectral shrinkage without residual quantization would help evaluate the proposed “spectral regularization” interpretation.

8. Expand the evaluation to at least one larger or architecturally different model, if computationally feasible, and clarify how well the method transfers across grouped-query attention, different head dimensions, and positional encoding variants.

9. Compare with additional recent KV-cache compression approaches under matched memory budgets and carefully include all auxiliary storage. If some methods cannot be compared directly because they modify attention execution or require retraining, this should be explained.

10. Improve the presentation of the algorithm and notation. In particular, it would help to provide a compact implementation-oriented description that clearly distinguishes singular values from eigenvalues, describes how the noise spectrum is imputed, and lists all hyperparameters and default values in one place.

11. Reframe claims about low-rank components as “shared context” unless supported by additional semantic analyses. At present, the evidence more directly establishes prominent low-rank structure than a specific semantic interpretation of that structure.

I view the work as promising and potentially suitable for TMLR after the central theoretical claims are clarified and the key ablations and practical-cost analyses are added.

---

### Review · Reviewer_oPmH · 2026-07-21

**Summary Of Contributions:**

This paper proposes eOptShrinkQ, a two-stage KV cache compression pipeline. The core idea is to model a block of key or value vectors as a spiked random matrix: a low-rank "shared context" signal plus a full-rank per-token residual with separable covariance. The low-rank part is extracted by eOptShrink (Su & Wu, 2025), an optimal singular value shrinkage method that handles colored noise and estimates the rank automatically via the BBP phase transition. The residual is then quantized row by row with TurboQuant's MSE-optimal scalar quantizer. The claimed benefit is that removing the low-rank structure restores the (approximate) isotropy that per-vector scalar quantization assumes, so no outlier handling, sub-grouping, or QJL bias correction is needed, and all bits go to reconstruction.

The paper provides asymptotic guarantees (residual spectrum matches the noise, inner product bias reduced by 1/(1+SNR), coordinate delocalization of residual rows), and evaluates on Llama-3.1-8B-Instruct and Ministral-8B-Instruct at three levels: per-head reconstruction metrics, LongBench (16 tasks), and multi-needle retrieval. Reported results show eOptShrinkQ at ~2.2 bits matching or beating TurboQuant at 3.0 bits and coming within 1.6-2.2 points of FP16 on LongBench.

Strengths: the conceptual framing is genuinely useful. The distinction from prior SVD-based KV compression (Palu, CSKV, SVDq, xKV), which uses SVD for dimensionality reduction, is real: here the residual stays full-dimensional and SVD acts as a preprocessing step that repairs the quantizer's input distribution. The choice of eOptShrink is also well matched to the problem (n and d comparable, non-white residual statistics, rank that varies across layers and heads). I checked the two main building blocks (TurboQuant and eOptShrink) and both are represented accurately. The layer-wise rank variation observation (Section 5) gives good support for why adaptive rank matters over fixed-rank SVD.

Weaknesses, elaborated below: the main baselines are evaluated in a weakened configuration; the headline "matches or exceeds FP16" claims rest on sub-1-point differences with no variance estimates; the theory is asymptotic and its constants are not controlled at the operating point d=128; and the "restores isotropy" story is stronger than what is actually proven, since the paper's own noise model has non-identity column covariance B.

**Audience:**

Yes

**Audience Explanation:**

KV cache compression is an active area with clear practical stakes, and the paper's framing is a real contribution to it: it identifies the low-rank shared structure as the root cause of the anisotropy that existing methods (per-channel quantization in KIVI, outlier handling, QJL correction) engineer around, and removes it at the source. The connection to the optimal shrinkage literature, in particular the colored-noise eOptShrink variant with automatic rank selection, is new in this application and will interest both the efficient-inference community and people working on random matrix methods looking for ML applications. The negative finding about QJL (bias-variance tradeoff hurting multi-needle retrieval) is also useful to practitioners independent of the main method. Even readers who end up unconvinced by the "exceeds FP16" regularization story will find the decomposition perspective worth knowing.

**Broader Impact Concerns:**

N/A.

**Claims And Evidence:**

Yes

**Claims Explanation:**

Most of the paper's claims are supported, but several of the central ones are not yet, and they are the ones in the abstract.

1. The baseline comparison is not apples-to-apples. Section 4 states that no sub-grouping or outlier handling is used for any method, "for fair comparison". But sub-grouping and outlier handling are part of TurboQuant's intended configuration. The paper's argument is precisely that eOptShrinkQ makes those tricks unnecessary; the honest test of that argument is eOptShrinkQ-without-tricks vs. TurboQuant-with-tricks. As reported, I cannot tell how much of the "nearly one bit saved" comes from the method and how much comes from handicapping the baseline.


2. The theoretical guarantees are asymptotic but presented as guarantees at d=128. Proposition 6 and Corollary 6 are stochastic dominance statements with rates like d^{-1/2} (about 0.09 at d=128) and additional phi_d terms. The appendix is honest about this (Remark A.2 explicitly says the guarantees hold "as approximations whose quality we verify empirically" for fixed inputs), but the abstract and introduction sell "three guarantees". The signal-vector randomness assumption A.2(v) is acknowledged not to hold at inference time. This is a presentation problem more than a correctness problem, but it needs fixing.


3. The h_t = local-context + token-specific decomposition (Eq. 1) is asserted, not verified. The spectral validation in Section 4.1 (outliers above a bulk edge) is consistent with the spiked model, but any matrix with a few large singular values produces the same picture. The spectra do not by themselves confirm the semantic story that the spike is "shared local context" and the bulk is "token-specific content", nor the independence of the two parts.

4. No wall-clock measurements. The overhead argument is purely asymptotic. eOptShrink runs a full SVD per 128x128 block, per head, per layer, per chunk; at 100K context that is hundreds of chunks times all layers and heads, and many small SVDs tend to be latency-bound rather than FLOP-bound on GPUs. "Negligible compared to attention" needs at least one measured prefill latency number.

None of these strike me as fatal. Points 1 need new experiments; points 2, 3, 4 can largely be addressed by more careful claims plus one or two targeted ablations.

**Requested Changes:**

Critical (needed to secure my recommendation):

1. Add a comparison against TurboQuant in its best configuration, with sub-grouping and outlier handling as in the original paper. If eOptShrinkQ still wins at fewer bits, the headline claim stands and becomes much stronger. If not, the claims need to be scoped to the uniform no-subgrouping setting.


2. Reframe the theory section: state clearly in the introduction and abstract that Proposition 6 / Corollary 6 are asymptotic results whose constants are not controlled at d=128, and that Assumption A.2(v) does not hold for a fixed input at inference time (the appendix already says this; the front of the paper should too).


Would strengthen the work:

1. Fix the broken references: the PolarQuant citation renders as "(? )" in Related Work.


2. One measured prefill latency comparison (with and without eOptShrinkQ) on a realistic long-context workload, and a note on decode-time reconstruction cost.


3. Some direct evidence for the decomposition story in Eq. 1 beyond the spectra, e.g. showing that the extracted S correlates with block-level semantic content (topic shift across blocks) while R does not, or that consecutive-token blocks show stronger low-rank structure than shuffled-token blocks (the paper predicts this via RoPE but never tests it).


4. Results at another head dimension (d=64) or a GQA/MLA-style cache, to show the approach is not tuned to d=128. Relatedly, a candid discussion of values, where the spike sits close to the bulk edge (Figures 3, 7) and rank estimation is presumably less stable.
Report the distribution of estimated ranks (not just the mean) and how often r_hat = 0 occurs, since the pipeline branches on it.